# Sample Efficient Stochastic Gradient Iterative Hard Thresholding Method for Stochastic Sparse Linear Regression with Limited Attribute Observation

**Tomoya Murata**
NTT DATA Mathematical Systems Inc. , Tokyo, Japan
`murata@msi.co.jp`

**Taiji Suzuki**
Department of Mathematical Informatics,
Graduate School of Information Science and Technology,
The University of Tokyo, Tokyo, Japan
Center for Advanced Intelligence Project, RIKEN, Tokyo, Japan
`taiji@mist.i.u-tokyo.ac.jp`

## Abstract

We develop new stochastic gradient methods for efficiently solving sparse linear regression in a partial attribute observation setting, where learners are only allowed to observe a fixed number of actively chosen attributes per example at training and prediction times. It is shown that the methods achieve essentially a sample complexity of $O(1/\varepsilon)$ to attain an error of $\varepsilon$ under a variant of restricted eigenvalue condition, and the rate has better dependency on the problem dimension than existing methods. Particularly, if the smallest magnitude of the non-zero components of the optimal solution is not too small, the rate of our proposed *Hybrid* algorithm can be boosted to near the minimax optimal sample complexity of *full information* algorithms. The core ideas are (i) efficient construction of an unbiased gradient estimator by the iterative usage of the hard thresholding operator for configuring an exploration algorithm; and (ii) an adaptive combination of the exploration and an exploitation algorithms for quickly identifying the support of the optimum and efficiently searching the optimal parameter in its support. Experimental results are presented to validate our theoretical findings and the superiority of our proposed methods.

## 1 Introduction

In real-world sequential prediction scenarios, the features (or attributes) of examples are typically high-dimensional and construction of the all features for each example may be expensive or impossible. One of the example of these scenarios arises in the context of medical diagnosis of a disease, where each attribute is the result of a medical test on a patient [4]. In this scenarios, observations of the all features for each patient may be impossible because it is undesirable to conduct the all medical tests on each patient due to its physical and mental burden.

In limited attribute observation settings [1, 4] , learners are only allowed to observe a given number of attributes per example at training time. Hence learners need to update their predictor based on the actively chosen attributes which possibly differ from example to example.

Several methods have been proposed to deal with this setting in linear regression problems. Cesa-Bianchi et al. [4] have proposed a generalized stochastic gradient descent algorithm [16, 5, 14] based

on the ideas of picking observed attributes randomly and constructing a noisy version of all attributes using them. Hazan and Koren [7] have proposed an algorithm combining a stochastic variant of *EG* algorithm [11] with the idea in [4], which improves the dependency of the problem dimension of the convergence rate proven in [4].

In these work, limited attribute observation settings only at training time have been considered. However, it is natural to assume that the observable number of attributes *at prediction time* is same as the one at training time. This assumption naturally requires the *sparsity* of output predictors.

Despite the importance of the requirement of the sparsity of predictors, a hardness result in this setting is known. Foster et al. [6] have considered online (agnostic) sparse linear regression in the limited attribute observation setting. They have shown that no algorithm running in polynomial time per example can achieve any sub-linear regret unless $\mathrm{NP} \subset \mathrm{BPP}$. Also it has been shown that this hardness result holds in stochastic i.i.d. (non-agnostic) settings in [8]. These hardness results suggest that some additional assumptions are needed.

More recently, Kale and Karnin [10] have proposed an algorithm based on *Dantzig Selector* [3], which run in polynomial time per example and achieve sub-linear regrets *under restricted isometry condition* [2], which is well-known in sparse recovery literature. Particularly in non-agnostic settings, the proposed algorithm achieves a sample complexity of $\widetilde{O}(1/\varepsilon)^1$, but the rate has bad dependency on the problem dimension. Additionally, this algorithm requires large memory cost since it needs to store the all observed samples due to the applications of Dantzig Selector to the updated design matrices. Independently, Ito et al. [8] have also proposed three efficient runtime algorithms based on *regularized dual averaging* [15] with their proposed exploration-exploitation strategies in non-agnostic settings under *linear independence of features* or *compatibility*[2]. The one of the three algorithms achieves a sample complexity of $O(1/\varepsilon^2)$ under linear independence of features, which is worse than the one in [10], but has better dependency on the problem dimension. The other two algorithms also achieve a sample complexity of $O(1/\varepsilon^2)$, but the additional term independent to $\varepsilon$ has unacceptable dependency on the problem dimension.

As mentioned above, there exist several efficient runtime algorithms which solve sparse linear regression problem with limited attribute observations under suitable conditions. However , the convergence rates of these algorithms have bad dependency on the problem dimension or on desired accuracy. Whether more efficient algorithms exist is a quite important and interesting question.

**Main contribution**    In this paper, we focus on stochastic i.i.d. (non-agnostic) sparse linear regression in the limited attribute observation setting and propose new sample efficient algorithms in this setting. The main feature of proposed algorithms is summarized as follows:

> Our algorithms achieve a sample complexity of $\widetilde{O}(1/\varepsilon)$ with much better dependency on the problem dimension than the ones in existing work. Particularly, if the smallest magnitude of the non-zero components of the optimal solution is not too small, the rate can be boosted to near the minimax optimal sample complexity of *full information* algorithms.

Additionally, our algorithms also possess run-time efficiency and memory efficiency, since the average run-time cost per example and the memory cost of the proposed algorithms are in order of the number of observed attributes per example and of the problem dimension respectively, that are better or comparable to the ones of existing methods.

We list the comparisons of our methods with several preceding methods in our setting in Table 1.

## 2    Notation and Problem Setting

In this section, we formally describe the problem to be considered in this paper and the assumptions for our theory.

[†]Note that the necessary number of observed attributes per example *at prediction time* is $s_*$, that is nearly same as the other algorithms in table 1.

Table 1: Comparisons of our methods with existing methods in our problem setting. *Sample complexity* means the necessary number of samples to attain an error $\varepsilon$. "*# of observed attrs per ex.*" indicates the necessary number of observed attributes per example at training time which the algorithm requires at least. $s'$ is the number of observed attributes per example, $s_*$ is the size of the support of the optimal solution, $d$ is the problem dimension, $\varepsilon$ is the desired accuracy and $r_{\min}^2$ is the smallest magnitude of the non-zero components of the optimal solution. We regard the smoothness and strong convexity parameters of the objectives derived from the additional assumptions and the boundedness parameter of the input data distribution as constants. $\widetilde{O}$ hides extra log-factors for simplifying the notation.

| | Sample complexity | # of observed attrs per ex. | Additional assumptions | Objective type |
|---|---|---|---|---|
| Dantzig [10] | $\widetilde{O}\left(\frac{ds_*^2}{s'}\left(\sigma + \frac{d}{s'}\right)^2 \frac{1}{\varepsilon}\right)$ | $1^\dagger$ | restricted isometry condition | Regret |
| RDA1 [8] | $O\left(d^2\frac{\sigma^2}{\varepsilon^2}\right)$ | $s_* + 2$ | linear independence of features | Regret |
| RDA2 [8] | $O\left(\frac{d^{16}}{s'^{16}} + d\frac{\sigma^2}{\varepsilon^2}\right)$ | $s_*$ | linear independence of features | Regret |
| RDA3 [8] | $O\left(\frac{d^{16}}{s'^{16}} + d\frac{\sigma^2}{\varepsilon^2}\right)$ | $s_*$ | compatibility | Regret |
| Exploration | $\widetilde{O}\left(\frac{ds_*^2}{s'} + \frac{ds_*}{s'}\frac{\sigma^2}{\varepsilon}\right)$ | $O(s_*)$ | restricted smoothness & restricted strong convexity | Expected risk |
| Hybrid | $\widetilde{O}\left(\frac{ds_*^2}{s'} + \left(\frac{s_*}{r_{\min}^2} \wedge \frac{ds_*}{s'}\right)\frac{\sigma^2}{\varepsilon}\right)$ | $O(s_*)$ | restricted smoothness & restricted strong convexity | Expected risk |

## 2.1 Notation

We use the following notation in this paper.

- $\|\cdot\|$ denotes the Euclidean $L_2$ norm $\|\cdot\|_2$: $\|x\| = \|x\|_2 = \sqrt{\sum_i x_i^2}$.
- For natural number $m, n$, $[m, n]$ denotes the set $\{m, m+1, \ldots, n\}$. If $m = 1$, $[1, n]$ is abbreviated as $[n]$.
- $\mathcal{H}_s$ denotes the projection onto $s$-sparse vectors, i.e., $\mathcal{H}_s(\theta') = \text{argmin}_{\theta\in\mathbb{R}^d, \|\theta\|_0 \leq s}\|\theta - \theta'\|$ for $s \in \mathbb{N}$, where $\|\theta\|_0$ denotes the number of non-zero elements of $\theta$.
- For $x \in \mathbb{R}^d$, $x|_j$ denotes the $j$-th element of $x$. For $S \subset [d]$, we use $x|_S \in \mathbb{R}^{|S|}$ to denote the restriction of $x$ to $S$: $(x|_S)|_j = x_j$ for $j \in S$.

## 2.2 Problem definition

In this paper, we consider the following sparse linear regression model:

$$y = \theta_*^\top x + \xi, \text{ where } \|\theta_*\|_0 = s_*, x \sim D_X, \tag{1}$$

where $\xi$ is a mean zero sub-Gaussian random variable with parameter $\sigma^2$, which is independent to $x \sim D_X$. We denote $D$ as the joint distribution of $x$ and $y$.

For finding true parameter $\theta_*$ of model (1), we focus on the following optimization problem:

$$\min_{\|\theta\|_0 \leq s, \theta\in\mathbb{R}^d} \{\mathcal{L}(\theta) \stackrel{\text{def}}{=} \mathbb{E}_{(x,y)\sim D}[\ell_y(\theta^\top x)]\}, \tag{2}$$

where $\ell_y(a)$ is the standard squared loss $(a - y)^2$ and $s \geq s_*$ is some integer. We can easily see that true parameter $\theta_*$ is an optimal solution of the problem (2).

**Limited attribute observation**  We assume that only a small subset of the attributes which we actively choose per example rather than all attributes can be observed at both training and prediction time. In this paper, we aim to construct algorithms which solve problem (2) with only observing $s'(\geq s \geq s_*) \in [d]$ attributes per example. Typically, the situation $s' \ll d$ is considered.

## 2.3 Assumptions

We make the following assumptions for our analysis.

**Assumption 1** (Boundedness of data). For $x \sim D_X$, $\|x\|_\infty \le R_\infty$ with probability one.

**Assumption 2** (Restricted smoothness of $\mathcal{L}$). Objective function $\mathcal{L}$ satisfies the following restricted smoothness condition:

$$\forall s \in [d], \exists L_s > 0, \forall \theta_1, \theta_2 \in \mathbb{R}^d : \|\theta_1\|_0, \|\theta_2\|_0 \le s \Rightarrow \mathcal{L}(\theta_1) \le \mathcal{L}(\theta_2) + \langle \nabla \mathcal{L}(\theta_2), \theta_1 - \theta_2 \rangle + \frac{L_s}{2}\|\theta_1 - \theta_2\|^2.$$

**Assumption 3** (Restricted strong convexity of $\mathcal{L}$). Objective function $\mathcal{L}$ satisfies the following restricted strong convexity condition:

$$\forall s \in [d], \exists \mu_s > 0, \forall \theta_1, \theta_2 \in \mathbb{R}^d : \|\theta_1\|_0, \|\theta_2\|_0 \le s \Rightarrow \mathcal{L}(\theta_2) + \langle \nabla \mathcal{L}(\theta_2), \theta_1 - \theta_2 \rangle + \frac{\mu_s}{2}\|\theta_1 - \theta_2\|^2 \le \mathcal{L}(\theta_1).$$

By the restricted strong convexity of $\mathcal{L}$, we can easily see that the true parameter of model (1) is the unique optimal solution of optimization problem (2). We denote the condition number $L_s/\mu_s$ by $\kappa_s$.

*Remark.* In linear regression settings, Assumptions 2 and 3 are equivalent to assuming

$$\forall s \in [d], \exists L_s > 0 : \sup_{\theta \in \mathbb{R}^d \setminus \{0\}, \|\theta\|_0 \le 2s} \frac{\theta^\top \mathbb{E}_{x \sim D_X}[xx^\top]\theta}{\|\theta\|^2} \le L_s$$

and

$$\forall s \in [d], \exists \mu_s > 0 : \inf_{\theta \in \mathbb{R}^d \setminus \{0\}, \|\theta\|_0 \le 2s} \frac{\theta^\top \mathbb{E}_{x \sim D_X}[xx^\top]\theta}{\|\theta\|^2} \ge \mu_s$$

respectively. Note that these conditions are stronger than *restricted eigenvalue condition*, but are weaker than *restricted isometry condition*.

# 3 Approach and Algorithm Description

In this section, we illustrate our main ideas and describe the proposed algorithms in detail.

## 3.1 Exploration algorithm

One of the difficulties in partial information settings is that the standard stochastic gradient is no more available. In linear regression settings, the gradient what we want to estimate is given by $\mathbb{E}_{(x,y)\sim D}[\ell_y'(\theta^\top x)x] = \mathbb{E}_{(x,y)\sim D}[2(\theta^\top x - y)x]$. In general, we need to construct unbiased estimators of $\mathbb{E}_{(x,y)\sim D}[yx]$ and $\mathbb{E}_{x \sim D_X}[xx^\top]$. A standard technique is an usage of $\hat{x}$, which is defined as $\hat{x}|_j = x|_j$ ($j \in S$) and $\hat{x}|_j = 0$ ($j \notin S$), where $S \subset [d]$ is randomly observed with $|S| = s'$ and $x \sim D_X$. Then we obtain an unbiased estimator of $\mathbb{E}_{(x,y)\sim D}[yx]$ as $y\frac{d}{s'}\hat{x}$. Similarly, an unbiased estimator of $\mathbb{E}_{x \sim D_X}[xx^\top]$ is given by $\hat{x}\hat{x}^\top$ with adequate element-wise scaling. Note that particularly the latter estimator has a quite large variance because the probability that the $(i,j)$- entry of $\hat{x}\hat{x}^\top$ becomes non-zero is $O(s'^2/d^2)$ when $i \ne j$, which is very small when $s' \ll d$.

If the updated solution $\theta$ is sparse, computing $\theta^\top x$ requires only observing the attributes of $x$ which correspond to the support of $\theta$ and there exists no need to estimate $\mathbb{E}_{x \sim D_X}[xx^\top]$, which has a potentially large variance. However, this idea is not applied to existing methods because they do not ensure the sparsity of the updated solutions at training time and generate sparse output solutions *only at prediction time* by using the hard thresholding operator.

Iterative applications of the hard thresholding to the updated solutions at training time ensure the sparsity of them and an efficient construction of unbiased gradient estimators is enabled. Also we can fully utilize the restricted smoothness and restricted strong convexity of the objective (Assumption 2 and 3) due to the sparsity of the updated solutions if the optimal solution of the objective is sufficiently sparse.

Now we present our proposed estimator. Motivated by the above discussion, we adopt the iterative usage of the hard thresholding at training time. Thanks to the usage of the hard thresholding operator that projects dense vectors to $s$-sparse ones, we are guaranteed that the updated solutions are $s(< s')$-sparse, where $s'$ is the number of observable attributes per example. Hence we can efficiently estimate $\mathbb{E}_{x \sim D_X}[\theta^\top xx]$ as $\theta^\top x\hat{x}$ with adequate scaling. As described above, computing $\theta^\top x$ can be efficiently

executed and only requires observing $s$ attributes of $x$. Thus an naive algorithm based on this idea becomes as follows:

$$\text{Sample } (x_t, y_t) \sim D.$$

Observe $x_t|_{\mathrm{supp}(\theta_{t-1}) \cup S}$, where $S$ is a random subset of $[d]$ with $|S| = s' - s$.

Compute $g_t = 2(\theta_{t-1}^\top x_t - y_t) \dfrac{d}{s' - s} \hat{x}_t$.

Update $\theta_t = \mathcal{H}_s(\theta_{t-1} - \eta_t g_t)$.

for $t = 1, 2, \ldots, T$. Unfortunately, this algorithm has no theoretical guarantee due to the use of the hard thresholding. Generally, stochastic gradient methods need to decrease the learning rate $\eta_t$ as $t \to \infty$ for reducing the noise effect caused by the randomness in the construction of gradient estimators. Then a large amount of stochastic gradients with small step sizes are cumulated for proper updates of solutions. However, the hard thresholding operator clears the cumulated effect on the outside of the support of the current solution at every update and thus the convergence of the above algorithm is not ensured if decreasing learning rate is used. For overcoming this problem, we adopt the standard mini-batch strategy for reducing the variance of the gradient estimator without decreasing the learning rate.

We provide the concrete procedure based on the above ideas in Algorithm 1. We sample $\lceil \frac{d}{s'-s} \rceil \times B_t$ examples per one update. The support of the current solution and deterministically selected $s' - s$ attributes are observed for each example. For constructing unbiased gradient estimator $g_t$, we average the $B_t$ unbiased gradient estimators, where each estimator is the concatenation of block-wise unbiased gradient estimators of $\lceil \frac{d}{s'-s} \rceil$ examples. Note that a constant step size is adopted. We call Algorithm 1 as *Exploration* since each coordinate is equally treated with respect to the construction of the gradient estimator.

### 3.2 Refinement of Algorithm 1 using exploitation and its adaptation

As we will state in Theorem 4.1 of Section 4, Exploration (Algorithm 1) achieves a linear convergence when adequate leaning rate, support size $s$ and mini-batch sizes $\{B_t\}_{t=1}^\infty$ are chosen. Using this fact, we can show that Algorithm 1 identifies the optimal support *in finite iterations* with high probability. When once we find the optimal support, it is much efficient to optimize the parameter on it rather than globally. We call this algorithm as *Exploitation* and describe the detail in Algorithm 2. Ideally, it is desirable that first we run Exploration (Algorithm 1) and if we find the optimal support, then we switch from Exploration to Exploitation (Algorithm 2). However, whether the optimal support has been found is uncheckable in practice and the theoretical number of updates for finding it depends on the smallest magnitude of the non-zero components of the optimal solution, which is unknown. Therefore, we need to construct an algorithm which combines Exploration and Exploitation, and is *adaptive* to the unknown value. We give this adaptive algorithm in Algorithm 3. This algorithm alternately uses Exploration and Exploitation. We can show that Algorithm 3 achieves at least the same convergence rate as Exploration, and thanks to the usage of Exploitation, its rate can be much boosted when the smallest magnitude of the non-zero components of the optimal solution is not too small. We call this algorithm as *Hybrid*.

## 4 Convergence Analysis

In this section, we provide the convergence analysis of our proposed algorithms. We use $\widetilde{O}$ notation to hide extra log-factors for simplifying the statements. Here, the log-factors have the form $O\left(\log\left(\frac{\kappa_s d}{\delta}\right)\right)$, where $\delta$ is a confidence parameter used in the statements.

### 4.1 Analysis of Algorithm 1

The following theorem implies that Algorithm 1 with sufficiently large mini-batch sizes $\{B_t\}_{t=1}^\infty$ achieves a linear convergence.

**Theorem 4.1** (Exploration). *Let $T \in \mathbb{N}$ and $\theta_0 \in \mathbb{R}^d$. For Algorithm 1, if we adequately choose $s = O\left(\kappa_s^2 s_*\right)$, $\eta = \Theta\left(\frac{1}{L_s}\right)$ and $\check{\alpha} = \Theta\left(\frac{1}{\kappa_s}\right)$, then for any $s'(> s) \in [d]$, $\delta \in (0, 1)$ and $\Delta > 0$*

**Algorithm 1:** Exploration($\theta_0 \in \mathbb{R}^d$, $\eta > 0$, $s', s \in [d]$ ($s' > s$), $\{B_t\}$, $T \in \mathbb{N}$)

Set $\theta_0 = \mathcal{H}_s(\theta_0)$, $d' = \left\lceil \frac{d}{s'-s} \right\rceil$ and $J_i = [(s'-s)(i-1)+1, (s'-s)i \wedge d]$ for $i \in [d']$.
**for** $t = 1$ to $T$ **do**
   Set $S_{t-1} = \mathrm{supp}(\theta_{t-1})$.
   Sample $\left(x_i^{(b)}, y_i^{(b)}\right) \sim D$ for $i \in [d']$ and $b \in [B_t]$.
   Observe $x_i^{(b)}|_{J_i}$, $x_i^{(b)}|_{S_{t-1}}$ and $y_i^{(b)}$ for $i \in [d']$ and $b \in [B_t]$.
   Compute $g_t|_{J_i} = \frac{1}{B_t} \sum_{b=1}^{B_t} \ell'_{y_i^{(b)}}(\theta_{t-1}|_{S_{t-1}}^\top x_i^{(b)}|_{S_{t-1}}) x_i^{(b)}|_{J_i}$ for $i \in [d']$.
   Update $\theta_t = \mathcal{H}_s(\theta_{t-1} - \eta g_t)$.
**end for**
**return** $\theta_T$.

---

**Algorithm 2:** Exploitation($\theta_0 \in \mathbb{R}^d$, $\eta > 0$, $\{B_t\}$, $T \in \mathbb{N}$)

Set $S_0 = \mathrm{supp}(\theta_0)$.
**for** $t = 1$ to $T$ **do**
   Sample $\left(x^{(b)}, y^{(b)}\right) \sim D$ for $b \in [B_t]$.
   Observe $x^{(b)}|_{S_0}$ and $y^{(b)}$ for $b \in [B_t]$.
   Compute $g_t|_{S_0} = \frac{1}{B_t} \sum_{b=1}^{B_t} \ell'_{y^{(b)}}(\theta_{t-1}|_{S_0}^\top x^{(b)}|_{S_0}) x^{(b)}|_{S_0}$.
   Set $g_t|_{S_0^{\complement}} = 0$.
   Update $\theta_t = \theta_{t-1} - \eta g_t$.
**end for**
**return** $\theta_T$.

---

*there exists* $B_t = \widetilde{O}\left(\kappa_s^2 \frac{R_\infty^4}{L_s^2} s^2 \vee \frac{\sigma^2}{\Delta} \frac{R_\infty^2}{L_s} \frac{Ts}{(1-\check{\alpha})^T}\right)$ $(t = 1, \ldots, \infty)$ *such that*

$$P\left(\mathcal{L}(\theta_T) - \mathcal{L}(\theta_*) \leq (1 - \check{\alpha})^T \left(\mathcal{L}(\theta_0) - \mathcal{L}(\theta_*) + \Delta\right)\right) \geq 1 - \delta.$$

The proof of Theorem 4.1 is found in Section A.1 of the supplementary material.

From Theorem 4.1, we obtain the following corollary, which gives a sample complexity of the algorithm.

**Corollary 4.2** (Exploration). *For Algorithm 1, under the settings of Theorem 4.1 with $\Delta = \mathcal{L}(\theta_0) - \mathcal{L}(\theta_*)$, the necessary number of observed samples to achieve $P(\mathcal{L}(\theta_T) - \mathcal{L}(\theta_*) \leq \varepsilon) \geq 1 - \delta$ is*

$$\widetilde{O}\left(\frac{\kappa_s R_\infty^4}{\mu_s^2} \frac{d s^2}{s' - s} + \frac{\kappa_s R_\infty^2}{\mu_s} \frac{ds}{s' - s} \frac{\sigma^2}{\varepsilon}\right).$$

The proof of Corollary 4.2 is given in Section A.2 of the supplementary material.

*Remark.* If we set $s' - s = \Theta(s)$ and assume that $\kappa_s$, $R_\infty^2$ and $\mu_s$ are $\Theta(1)$, Corollary 4.2 gives the sample complexity of $\widetilde{O}(ds_* + d\sigma^2/\varepsilon)$.

*Remark.* Corollary 4.2 implies that in full information settings, i.e., $s' - s = \Theta(d)$, Algorithm 4.2 achieves a sample complexity of $\widetilde{O}(s_*^2 + s_*\sigma^2/\varepsilon)$, if $\kappa_s$, $R_\infty^2$ and $\mu_s$ are regard as $\Theta(1)$. This rate is near the minimax optimal sample complexity of $\widetilde{O}(s_*\sigma^2/\varepsilon)$ in full information settings [13].

*Remark.* The estimator $\theta_T$ is guaranteed to be asymptotically consistent, because it can be easily seen that $\|\theta_T - \theta_*\|^2$ converges to 0 as $T \to \infty$ by using the restricted strong convexity of the objective $\mathcal{L}$ and its convergence rate is nearly same as the one of the objective gap $\mathcal{L}(\theta_T) - \mathcal{L}(\theta_*)$.

## 4.2 Analysis of Algorithm 2

Generally, Algorithm 2 does not ensure its convergence. However, the following theorem shows that running Algorithm 2 with sufficiently large batch sizes will not increase the objective values too

---
**Algorithm 3:** Hybrid($\widetilde{\theta}_0 \in \mathbb{R}^d$, $\eta > 0$, $s'$, $s \in [d]$ ($s' > s$), $\{B_{t,k}^-\}$, $\{B_{t,k}\}$, $\{T_k^-\}$, $\{T_k\}$, $K \in \mathbb{N}$)

   **for** $k = 1$ to $K$ **do**
      Update $\widetilde{\theta}_k^- = \text{Exploration}(\widetilde{\theta}_{k-1}, \eta, s', s, \{B_{t,k}^-\}_{t=1}^\infty, T_k^-)$.
      Update $\widetilde{\theta}_k = \text{Exploitation}(\widetilde{\theta}_k^-, \eta, \{B_{t,k}\}_{t=1}^\infty, T_k)$.
   **end for**
   **return** $\widetilde{\theta}_K$.
---

much. Moreover, if the support of the optimal solution is included in the one of a initial point, then Algorithm 2 also achieves a linear convergence.

**Theorem 4.3** (Exploitation). *Let* $T \in \mathbb{N}$, $\theta_0 \in \mathbb{R}^d$ *and* $s \geq |\text{supp}(\theta_0)| \vee |\text{supp}(\theta_*)| \in \mathbb{N}$. *For Algorithm 2, if we adequately choose* $\eta = \Theta\left(\frac{1}{L_s}\right)$ *and* $\check{\alpha} = \Theta\left(\frac{1}{\kappa_s}\right)$, *then for any* $\delta \in (0, 1)$ *and* $\Delta > 0$, *there exists* $B_t = \widetilde{O}\left(\frac{R_\infty^4}{\mu_s^2} T s^2 \vee \frac{\sigma^2}{\Delta} \frac{R_\infty^2}{L_s} \frac{Ts}{(1-\check{\alpha})^T}\right)$ $(t = 1, \ldots, \infty)$ *such that*

$$
\begin{cases}
P\left(\mathcal{L}(\theta_T) - \mathcal{L}(\theta_*) \leq \frac{1}{1-\check{\alpha}}(\mathcal{L}(\theta_0) - \mathcal{L}(\theta_*)) + \Delta\right) \geq 1 - \delta & \textit{(Generally)}, \\
P\left(\mathcal{L}(\theta_T) - \mathcal{L}(\theta_*) \leq (1 - \check{\alpha})^T (\mathcal{L}(\theta_0) - \mathcal{L}(\theta_*) + \Delta)\right) \geq 1 - \delta & \textit{(If } \text{supp}(\theta_*) \subset \text{supp}(\theta_0)) .
\end{cases}
$$

The proof of Theorem 4.3 is found in Section B of the supplementary material.

### 4.3 Analysis of Algorithm 3

Combining Theorem 4.1 and Theorem 4.3, we obtain the following theorem and corollary. These imply that using the adequate numbers of inner loops $\{T_k^-\}$, $\{T_k\}$ and mini-batch sizes $\{B_{t,k}^-\}$, $\{B_{t,k}\}$ of Algorithm 1 and Algorithm 2 respectively, Algorithm 3 is guaranteed to achieve the same sample complexity as the one of Algorithm 1 *at least*. Furthermore, if the smallest magnitude of the non-zero components of the optimal solution is not too small, its sample complexity can be much reduced.

**Theorem 4.4** (Hybrid). *We denote* $r_{\min} = \min_{j \in \text{supp}(\theta_*)} |\theta_*|_j|$ *and* $B_k(T, s, \check{\alpha}) = \kappa_s^2 \frac{R_\infty^4}{L_s^2} T s^2 \vee \frac{\sigma^2}{\check{\alpha}(1-\check{\alpha})^k (\mathcal{L}(\widetilde{\theta}_0) - \mathcal{L}(\theta_*))} \frac{R_\infty^2}{L_s} \frac{Ts}{(1-\check{\alpha})^T}$. *Let* $K \in \mathbb{N}$ *and* $\widetilde{\theta}_0 \in \mathbb{R}^d$. *If we adequately choose* $s = O(\kappa_s^2 s_*)$, $\eta = O\left(\frac{1}{L_s}\right)$ *and* $\check{\alpha} = \Theta\left(\frac{1}{\kappa_s}\right)$, *for any* $s'(> s) \in [d]$ *and* $\delta \in (0, \frac{1}{3})$, *Algorithm 3 with* $T_k^- = 3$, *and adequate* $T_k = \widetilde{T} = \left\lceil \frac{1}{\log((1-\check{\alpha}_s)^{-1})} \log\left(\frac{d}{\Theta(\kappa_s^2)(s'-s)} \vee 1\right)\right\rceil$, $B_{t,k}^- = \widetilde{O}(B_k(T_k^-, s, \check{\alpha}))$ *and* $B_{t,k} = \widetilde{O}(B_k(T_k, s, \check{\alpha}))$ *satisfies*

$$
\begin{cases}
P\left(\mathcal{L}(\widetilde{\theta}_K) - \mathcal{L}(\theta_*) \leq 2(1 - \check{\alpha})^K (\mathcal{L}(\widetilde{\theta}_0) - \mathcal{L}(\theta_*)\right) \geq 1 - \delta & \textit{(Generally)}, \\
P\left(\mathcal{L}(\widetilde{\theta}_K) - \mathcal{L}(\theta_*) \leq 2(1 - \check{\alpha})^{K+\widetilde{T}} (\mathcal{L}(\widetilde{\theta}_0) - \mathcal{L}(\theta_*))\right) \geq 1 - 2\delta & \textit{(if } K \geq \check{k} + 1),
\end{cases}
$$

*where* $\check{k} = \left\lceil \frac{1}{\log((1-\check{\alpha})^{-1})} \log\left(\frac{4(\mathcal{L}(\widetilde{\theta}_0) - \mathcal{L}(\theta_*))}{r_{\min}^2 \mu_s}\right)\right\rceil$.

The proof of Theorem 4.4 is found in Section C.1 of the supplementary material.

**Corollary 4.5** (Hybrid). *Under the settings of Theorem 4.5, the necessary number of observed samples to achieve* $P(\mathcal{L}(\widetilde{\theta}_K) - \mathcal{L}(\theta_*) \leq \varepsilon) \geq 1 - \delta$ *for Algorithm 3 is*

$$
\widetilde{O}\left(\frac{\kappa_s^3 R_\infty^4}{\mu_s^2} s^2 + \frac{\kappa_s R_\infty^4}{\mu_s^2} \frac{ds^2}{s' - s} + \frac{\kappa_s R_\infty^2}{\mu_s} \left(\frac{\kappa_s^2 s}{\mu_s r_{\min}^2} \wedge \frac{ds}{s' - s}\right) \frac{\sigma^2}{\varepsilon}\right).
$$

The proof of Corollary 4.5 is given in Section C.2 of the supplementary material.

*Remark.* From Corollary 4.5, if $\kappa_s^2/(\mu_s r_{\min}^2) \ll d/(s' - s)$, the sample complexity of Hybrid can be much better than the one of Exploration only. Particularly, if we assume that $\kappa_s$, $R_\infty/\mu_s$ and $\mu_s r_{\min}^2$ are $\Theta(1)$ and $s' - s = \Theta(s)$, Algorithm 3 achieves a sample complexity of $\widetilde{O}(ds_* + s_*\sigma^2/\varepsilon)$, which is asymptotically near the minimax optimal sample complexity of full information algorithms *even in partial information settings*. In this case, the complexity is significantly smaller than $\widetilde{O}(ds_* + d\sigma^2/\varepsilon)$ of Algorithm 1 in this situation.

# 5   Relation to Existing Work

In this section, we describe the relation between our methods and the most relevant existing methods. The methods of [4] and [7] solve the stochastic linear regression with limited attribute observation, but the limited information setting is only assumed at training time and not at prediction time, which is different from ours. Also their theoretical sample complexities are $O(1/\varepsilon^2)$ which is worse than ours. The method of [10] solve the sparse linear regression with limited information based on Dantzig Selector. It has been shown that the method achieves sub-linear regret in both agnostic (online) and non-agnostic (stochastic) settings under an online variant of restricted isometry condition. The convergence rate in non-agnostic cases is much worse than the ones of ours in terms of the dependency on the problem dimension $d$, but the method has high versatility since it has theoretical guarantees also in agnostic settings, which have not been focused in our work. The methods of [8] are based on regularized dual averaging with their exploration-exploitation strategies and achieve a sample complexity of $O(1/\varepsilon^2)$ under linear independence of features or compatibility, which is worse than $\widetilde{O}(1/\varepsilon)$ of ours. Also the rate of Algorithm 1 in [8] has worse dependency on the dimension $d$ than the ones of ours. Additionally theoretical analysis of the method assumes linear independence of features, which is much stronger than restricted isometry condition or our restricted smoothness and strong convexity conditions. The rate of Algorithm 2, 3 in [8] has an additional term which has quite terrible dependency on $d$, though it is independent to $\varepsilon$. Their exploration-exploitation idea is different from ours. Roughly speaking, these methods observe $s_*$ attributes which correspond to the coordinates that have large magnitude of the updated solution, and $s' - s_*$ attributes uniformly at random. This means that exploration and exploitation are combined in single updates. In contrast, our proposed Hybrid updates a predictor alternatively using Exploration and Exploitation. This is a big difference: if their scheme is adopted, the variance of the gradient estimator on the coordinates that have large magnitude of the updated solution becomes small, however the variance reduction effect is buried in the large noise derived from the other coordinates, and this makes efficient exploitation impossible. In [9] and [12], (stochastic) gradient iterative hard thresholding methods for solving empirical risk minimization with sparse constraints in full information settings have been proposed. Our Exploration algorithm can be regard as generalization of these methods to limited information settings.

# 6   Numerical Experiments

In this section, we provide numerical experiments to demonstrate the performance of the proposed algorithms through synthetic data and real data.

We compare our proposed Exploration and Hybrid with state-of-the-art Dantzig [10] and RDA (Algorithm 1[3]in [8]) in our limited attribute observation setting on a synthetic and real dataset. We randomly split the dataset into training (90%) and testing (10%) set and then we trained each algorithm on the training set and executed the mean squared error on the test set. We independently repeated the experiments 5 times and averaged the mean squared error. For each algorithm, we appropriately tuned the hyper-parameters and selected the ones with the lowest mean squared error.

**Synthetic dataset**   Here we compare the performances in synthetic data. We generated $n = 10^5$ samples with dimension $d = 500$. Each feature was generated from an i.i.d. standard normal. The optimal predictor was constructed as follows: $\theta_*|_j = 1$ for $j \in [13]$, $\theta_*|_j = -1$ for $j \in [14, 25]$ and $\theta_*|_j = 0$ for the other $j$. The optimal predictor has only 25 non-zero components and thus $s_* = 25$. The output was generated as $y = \theta_*^\top x + \xi$, where $\xi$ was generated from an i.i.d. standard normal. We set the number of observed attributes per example $s'$ as 50. Figure 1 shows the averaged mean squared error as a function of the number of observed samples. The error bars depict two standard deviation of the measurements. Our proposed Hybrid and Exploration outperformed the other two methods. RDA initially performed well, but its convergence slowed down. Dantzig showed worse performance than all the other methods. Hybrid performed better than Exploration and showed rapid convergence.

**Real dataset** Finally, we show the experimental results on a real dataset *CT-slice*[4]. CT-slice dataset consists of $n = 53,500$ CT images with $d = 383$ features. The target variable of each image denotes the relative location of the image on the axial axis. We set the number of observable attributes per example $s'$ as 20. In figure 2, the mean squared error is depicted against the number of observed examples. The error bars show two standard deviation of the measurements. Again, our proposed methods surpasses the performances of the existing methods. Particularly, the convergence of Hybrid was significantly fast and stable. In this dataset, Dantzig showed nice convergence and comparable to our Exploration. The convergence of RDA was quite slow and a bit unstable.

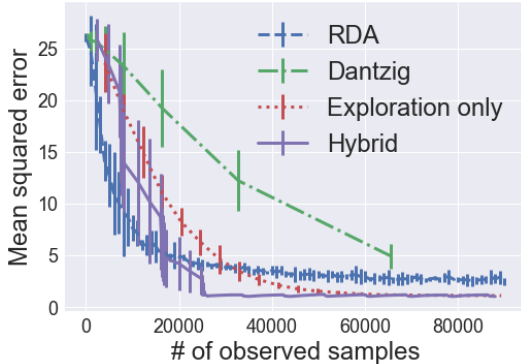

Figure 1: Comparison on synthetic data.

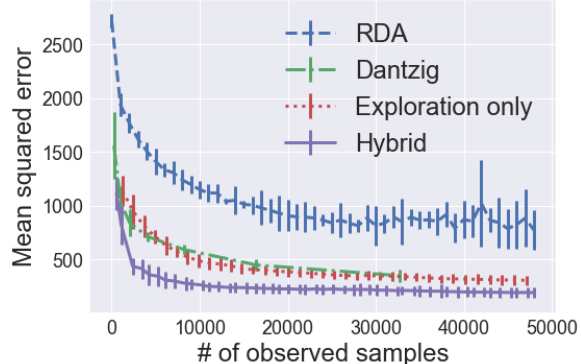

Figure 2: Comparison on CT-slice data.

## 7 Conclusion

We presented sample efficient algorithms for stochastic sparse linear regression problem with limited attribute observation. We developed *Exploration* algorithm based on an efficient construction of an unbiased gradient estimator by taking advantage of the iterative usage of hard thresholding in the updates of predictors . Also we refined Exploration by adaptively combining it with *Exploitation* and proposed *Hybrid* algorithm. We have shown that Exploration and Hybrid achieve a sample complexity of $\widetilde{O}(1/\varepsilon)$ with much better dependency on the problem dimension than the ones in existing work. Particularly, if the smallest magnitude of the non-zero components of the optimal solution is not too small, the rate of Hybrid can be boosted to near the minimax optimal sample complexity of full information algorithms. In numerical experiments, our methods showed superior convergence behaviors compared to preceding methods on synthetic and real data sets.

## Acknowledgement

TS was partially supported by MEXT Kakenhi (25730013, 25120012, 26280009, 15H05707 and 18H03201), Japan Digital Design, and JST-CREST.

## Footnotes

[1]$\widetilde{O}$ hides extra log-factors.

[3]In [8], three algorithms have been proposed (Algorithm 1, 2 and 3). We did not implement the latter two ones because the theoretical sample complexity of these algorithms makes no sense unless $d/s'$ is quite small due to the existence of the additional term $d^{16}/s'^{16}$ in it.

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
