[Supplementary Material]

# Supplementary material:
# Sample Efficient Stochastic Gradient Iterative Hard Thresholding Method for Stochastic Sparse Linear Regression with Limited Attribute Observation

**Tomoya Murata**
NTT DATA Mathematical Systems Inc. , Tokyo, Japan
murata@msi.co.jp

**Taiji Suzuki**
Department of Mathematical Informatics
Graduate School of Information Science and Technology, The University of Tokyo, Tokyo, Japan
Center for Advanced Intelligence Project, RIKEN, Tokyo, Japan
taiji@mist.i.u-tokyo.ac.jp

In this supplementary material, we provide the proofs of Theorem 4.1 and Corollary 4.2 (Section A), the one of Theorem 4.3 (Section B) and the ones of Theorem 4.4 and Corollary 4.5 (Section C).

## A  Analysis of Algorithm 1

In this section, we give the comprehensive proofs of Theorem 4.1 and Corollary 4.2.

### A.1  Proof of Theorem 4.1

The proof is essentially generalization of the one in [1] to stochastic and partial information settings.

**Proposition A.1** (Exploration)**.** *Suppose that* $\eta < \frac{1}{2L_s}$, $s \geq \max\left\{ 2\left(\frac{\frac{1}{4}+\frac{\eta L_s}{2}}{\frac{1}{4}-\frac{\eta L_s}{2}}\right) - 1, \frac{64}{\eta^2 \mu_s^2} + 1 \right\} s_*$
*and* $B_t \geq \frac{4s}{s_*}(s+s_*)^2 \frac{\left(\frac{5}{2}+\eta L_s\right)\eta \log\left(\frac{2d}{\delta_t}\right)}{\left(\frac{1}{4\eta}+\frac{L_s}{2}\right)}$. *Then for any* $\delta_t \in \left(0, \frac{1}{2}\right)$, *Algorithm 1 satisfies*

$$\mathcal{L}(\theta_t) - \mathcal{L}(\theta_*) \leq (1-\alpha)(\mathcal{L}(\theta_{t-1}) - \mathcal{L}(\theta_*)) + \frac{c_t}{B_t}$$

*with probability* $\geq 1 - 2\delta_t$, *where* $\alpha = \frac{1}{2}\left(1 - \frac{2s_*}{s+s_*}\right)\mu_s\left(\frac{1}{4} + \frac{\eta L_s}{2}\right)\eta$ *and* $c_t = 4\sigma^2 s\left(\frac{5}{2} + \eta L_s\right)\eta \log\left(\frac{d}{\delta_t}\right)$.

*Proof.* We denote $S_t$, $S_{t-1}$ and $S_*$ as $\mathrm{supp}(\theta_t)$, $\mathrm{supp}(\theta_{t-1})$ and $\mathrm{supp}(\theta_*)$ respectively. Also we define $\widetilde{S}_t = S_t \cup S_{t-1} \cup S_*$.
Since $\theta_t$ and $\theta_{t-1}$ are $s$-sparse, restricted smoothness of $\mathcal{L}$ implies

$$\mathcal{L}(\theta_t) - \mathcal{L}(\theta_{t-1}) \leq \langle \nabla\mathcal{L}(\theta_{t-1}), \theta_t - \theta_{t-1}\rangle + \frac{L_s}{2}\|\theta_t - \theta_{t-1}\|^2$$

$$= \langle g_t, \theta_t - \theta_{t-1}\rangle + \frac{L_s}{2}\|\theta_t - \theta_{t-1}\|^2 - \langle g_t - \nabla\mathcal{L}(\theta_{t-1}), \theta_t - \theta_{t-1}\rangle$$

Using Cauchy-Schwarz inequality and Young's inequality, we have

$$-\langle g_t - \nabla\mathcal{L}(\theta_{t-1}), \theta_t - \theta_{t-1}\rangle \leq \frac{1}{4\eta}\|\theta_t - \theta_{t-1}\|^2 + \eta\|g_t|_{S_t \cup S_{t-1}} - \nabla_{S_t \cup S_{t-1}}\mathcal{L}(\theta_{t-1})\|^2$$

$$\leq \frac{1}{4\eta}\|\theta_t - \theta_{t-1}\|^2 + 2s\eta\|g_t - \nabla\mathcal{L}(\theta_{t-1})\|_\infty^2.$$

Here the second inequality follows from the fact $|S_t \cup S_{t-1}| \leq 2s$.

Thus we get

$$\begin{aligned}
\mathcal{L}(\theta_t) - \mathcal{L}(\theta_{t-1}) \leq&\ \langle g_t, \theta_t - \theta_{t-1}\rangle + \left(\frac{1}{4\eta} + \frac{L_s}{2}\right)\|\theta_t - \theta_{t-1}\|^2 + 2s\eta\|g_t - \nabla\mathcal{L}(\theta_{t-1})\|_\infty^2 \\
=&\ \left(\frac{1}{4\eta} + \frac{L_s}{2}\right)\left(\|\theta_t - \theta_{t-1} + \eta g_t|_{\widetilde{S}_t}\|^2 - \eta^2\|g_t|_{\widetilde{S}_t}\|^2\right) \\
&+ \left(\frac{1}{2} - \eta L_s\right)\langle g_t, \theta_t - \theta_{t-1}\rangle + 2s\eta\|g_t - \nabla\mathcal{L}(\theta_{t-1})\|_\infty^2.
\end{aligned} \tag{1}$$

Also we have

$$\begin{aligned}
\langle g_t, \theta_t - \theta_{t-1}\rangle =&\ -\langle g_t|_{S_{t-1}\setminus S_t}, \theta_{t-1}|_{S_{t-1}\setminus S_t}\rangle - \eta\|g_t|_{S_t}\|^2 \\
=&\ -\langle g_t|_{S_{t-1}\setminus S_t}, \theta_{t-1}|_{S_{t-1}\setminus S_t} - \eta g_t|_{S_{t-1}\setminus S_t}\rangle - \eta\|g_t|_{S_{t-1}\setminus S_t}\|^2 - \eta\|g_t|_{S_t}\|^2 \\
\leq&\ \|g_t|_{S_{t-1}\setminus S_t}\|\|\theta_{t-1}|_{S_{t-1}\setminus S_t} - \eta g_t|_{S_{t-1}\setminus S_t}\| - \eta\|g_t|_{S_{t-1}\setminus S_t}\|^2 - \eta\|g_t|_{S_t}\|^2 \\
\leq&\ \frac{1}{2\eta}\|\theta_{t-1}|_{S_{t-1}\setminus S_t} - \eta g_t|_{S_{t-1}\setminus S_t}\|^2 - \frac{\eta}{2}\|g_t|_{S_{t-1}\setminus S_t}\|^2 - \eta\|g_t|_{S_t}\|^2 \\
\leq&\ \frac{1}{2\eta}\|\theta_{t-1}|_{S_t\setminus S_{t-1}} - \eta g_t|_{S_t\setminus S_{t-1}}\|^2 - \frac{\eta}{2}\|g_t|_{S_{t-1}\setminus S_t}\|^2 - \eta\|g_t|_{S_t}\|^2 \\
=&\ \frac{\eta}{2}\|g_t|_{S_t\setminus S_{t-1}}\|^2 - \frac{\eta}{2}\|g_t|_{S_{t-1}\setminus S_t}\|^2 - \eta\|g_t|_{S_t}\|^2 \\
\leq&\ -\frac{\eta}{2}\|g_t|_{S_t\cup S_{t-1}}\|^2.
\end{aligned}$$

Here, the second inequality is due to Cauchy-Schwartz inequality. The third inequality follows from the fact that $|S_{t-1}\setminus S_t| = |S_t\setminus S_{t-1}|$ and the definition of hard thresholding operator. The third equality is by the fact that $\theta_{t-1}|_{S_t\setminus S_{t-1}} = 0$.

If $1/2 - \eta L_s > 0$ is assumed, combing (1) with this fact yields

$$\begin{aligned}
\mathcal{L}(\theta_t) - \mathcal{L}(\theta_{t-1}) \leq&\ \left(\frac{1}{4\eta} + \frac{L_s}{2}\right)\left(\|\theta_t - \theta_{t-1} + \eta g_t|_{\widetilde{S}_t}\|^2 - \eta^2\|g_t|_{\widetilde{S}_t}\|^2\right) \\
&- \left(\frac{1}{2} - \eta L_s\right)\frac{\eta}{2}\|g_t|_{S_t\cup S_{t-1}}\|^2 + 2s\eta\|g_t - \nabla\mathcal{L}(\theta_{t-1})\|_\infty^2 \\
=&\ \left(\frac{1}{4\eta} + \frac{L_s}{2}\right)\left(\|\theta_t - \theta_{t-1} + \eta g_t|_{\widetilde{S}_t}\|^2 - \eta^2\|g_t|_{\widetilde{S}_t\setminus(S_{t-1}\cup S_*)}\|^2\right) \\
&- \left(\frac{1}{2} + \eta L_s\right)\frac{\eta}{2}\|g_t|_{S_{t-1}\cup S_*}\|^2 - \left(\frac{1}{2} - \eta L_s\right)\frac{\eta}{2}\|g_t|_{S_t\cup S_{t-1}}\|^2 \\
&+ 2s\eta\|g_t - \nabla\mathcal{L}(\theta_{t-1})\|_\infty^2.
\end{aligned} \tag{2}$$

Now we bound $\|\theta_t - \theta_{t-1} + \eta g_t|_{\widetilde{S}_t}\|^2 - \eta^2\|g_t|_{\widetilde{S}_t\setminus(S_{t-1}\cup S_*)}\|^2$. We need the following three lemmas.

*Lemma* A.2.
$$S_t \setminus S_{t-1} = S_t \setminus (S_{t-1} \cup S_*) \oplus (S_t \cap S_*) \setminus S_{t-1}.$$

*Lemma* A.3.
$$\exists R \subset S_{t-1} \setminus S_t : |R| = |S_t \setminus (S_{t-1} \cup S_*)|.$$

*Proof.* Note that $|S_{t-1} \setminus S_t| = |S_t \setminus S_{t-1}|$ since $|S_t| = |S_{t-1}|$. Thus by Lemma A.2, we have

$$|S_{t-1} \setminus S_t| = |S_t \setminus S_{t-1}| = |S_t \setminus (S_{t-1} \cup S_*)| + |(S_t \cap S_*) \setminus S_{t-1}| \geq |S_t \setminus (S_{t-1} \cup S_*)|.$$

This gives the desired result. $\qquad\square$

**Lemma A.4.** *For any $\widetilde{S} \subset [d]$, $\theta, \theta_* \in \mathbb{R}^{|\widetilde{S}|}$ and $s, s_*$ such that $s_* \leq s < |\widetilde{S}|$, if $\|\theta_*\|_0 \leq s_*$, it follows that*

$$\|\mathcal{H}_s(\theta) - \theta\|^2 \leq \frac{|\widetilde{S}| - s}{|\widetilde{S}| - s_*} \|\theta - \theta_*\|^2.$$

The proof is found in [1].

Observe that

$$
\begin{aligned}
\eta^2 \|g_t|_{\widetilde{S}_t \setminus (S_{t-1} \cup S_*)}\|^2 &= \|\theta_t|_{\widetilde{S}_t \setminus (S_{t-1} \cup S_*)}\|^2 \\
&= \|\theta_t|_{S_t \setminus (S_{t-1} \cup S_*)}\|^2 \\
&= \|\theta_{t-1}|_{S_t \setminus (S_{t-1} \cup S_*)} - \eta g_t|_{S_t \setminus (S_{t-1} \cup S_*)}\|^2 \\
&\geq \|\theta_{t-1}|_R - \eta g_t|_R\|^2 \\
&= \|\theta_t|_R - \theta_{t-1}|_R - \eta g_t|_R\|^2.
\end{aligned}
$$

Here the first inequality is due to the definition of hard thresholding operator. The last equality follows from the fact that $R \subset S_{t-1} \setminus S_t$.

Hence we get

$$
\begin{aligned}
&\|\theta_t - \theta_{t-1} + \eta g_t|_{\widetilde{S}_t}\|^2 - \eta^2 \|g_t|_{\widetilde{S}_t \setminus (S_{t-1} \cup S_*)}\|^2 \\
&= \|\theta_t|_{\widetilde{S}_t} - \theta_{t-1}|_{\widetilde{S}_t} + \eta g_t|_{\widetilde{S}_t}\|^2 - \eta^2 \|g_t|_{\widetilde{S}_t \setminus (S_{t-1} \cup S_*)}\|^2 \\
&\leq \|\theta_t|_{\widetilde{S}_t \setminus R} - \theta_{t-1}|_{\widetilde{S}_t \setminus R} + \eta g_t|_{\widetilde{S}_t \setminus R}\|^2. \\
&= \|\mathcal{H}_s(\theta_{t-1}|_{\widetilde{S}_t \setminus R} - \eta g_t|_{\widetilde{S}_t \setminus R}) - \theta_{t-1}|_{\widetilde{S}_t \setminus R} + \eta g_t|_{\widetilde{S}_t \setminus R}\|^2.
\end{aligned}
$$

Here the second equality is due to the fact that $S_t \subset \widetilde{S}_t \setminus R$. If $|\widetilde{S}_t \setminus R| \leq s$, then $\|\mathcal{H}_s(\theta_{t-1}|_{\widetilde{S}_t \setminus R} - \eta g_t|_{\widetilde{S}_t \setminus R}) - \theta_{t-1}|_{\widetilde{S}_t \setminus R} + \eta g_t|_{\widetilde{S}_t \setminus R}\|^2 = 0$. Thus, we can assume that $|\widetilde{S}_t \setminus R| > s$. From Lemma A.4, we have

$$
\begin{aligned}
&\|\mathcal{H}_s(\theta_{t-1}|_{\widetilde{S}_t \setminus R} - \eta g_t|_{\widetilde{S}_t \setminus R}) - \theta_{t-1}|_{\widetilde{S}_t \setminus R} + \eta g_t|_{\widetilde{S}_t \setminus R}\|^2 \\
&\leq \frac{|\widetilde{S}_t \setminus R| - s}{|\widetilde{S}_t \setminus R| - s_*} \|\theta_*|_{\widetilde{S}_t \setminus R} - \theta_{t-1}|_{\widetilde{S}_t \setminus R} + \eta g_t|_{\widetilde{S}_t \setminus R}\|^2 \\
&\leq \frac{|\widetilde{S}_t \setminus R| - s}{|\widetilde{S}_t \setminus R| - s_*} \|\theta_*|_{\widetilde{S}_t} - \theta_{t-1}|_{\widetilde{S}_t} + \eta g_t|_{\widetilde{S}_t}\|^2. \qquad (3)
\end{aligned}
$$

Observe that

$$
\begin{aligned}
|\widetilde{S}_t \setminus R| &\leq |S_t| + |(S_{t-1} \setminus S_t) \setminus R| + |S_*| \\
&= |S_t| + |S_{t-1} \setminus S_t| - |R| + |S_*| \\
&= |S_t| + |S_t \setminus S_{t-1}| - |R| + |S_*| \\
&= |S_t| + |S_t \setminus S_{t-1}| - |S_t \setminus (S_{t-1} \cup S_*)| + |S_*| \\
&= |S_t| + |S_t \cap S_* \setminus S_{t-1}| + |S_*| \\
&\leq s + s_* + s_* \\
&= s + 2s_*,
\end{aligned}
$$

Noting that $(x - b)/(x - a) \leq (x^+ - b)/(x^+ - a)$ for any $x \leq x^+$ and $a \leq b < x$ and applying the above inequality to (3) yield

$$
\begin{aligned}
&\|\mathcal{H}_s(\theta_{t-1}|_{\widetilde{S}_t \setminus R} - \eta g_t|_{\widetilde{S}_t \setminus R}) - \theta_{t-1}|_{\widetilde{S}_t \setminus R} + \eta g_t|_{\widetilde{S}_t \setminus R}\|^2 \\
&\leq \frac{2s_*}{s + s_*} \|\theta_*|_{\widetilde{S}_t} - \theta_{t-1}|_{\widetilde{S}_t} + \eta g_t|_{\widetilde{S}_t}\|^2.
\end{aligned}
$$

Therefore we get

$$\|\theta_t - \theta_{t-1} + \eta g_t|_{\widetilde{S}_t}\|^2 - \eta^2 \|g_t|_{\widetilde{S}_t \setminus (S_{t-1} \cup S_*)}\|^2$$

$$\leq \frac{2s_*}{s + s_*} \|\theta_*|_{\widetilde{S}_t} - \theta_{t-1}|_{\widetilde{S}_t} + \eta g_t|_{\widetilde{S}_t}\|^2$$

$$= \frac{2s_*}{s + s_*} \left( \|\theta_{t-1} - \theta_*\|^2 + 2\eta \langle g_t, \theta_* - \theta_{t-1} \rangle + \eta^2 \|g_t|_{\widetilde{S}_t}\|^2 \right)$$

$$= \frac{2s_*}{s + s_*} \left( \|\theta_{t-1} - \theta_*\|^2 + 2\eta \langle \nabla \mathcal{L}(\theta_{t-1}), \theta_* - \theta_{t-1} \rangle + \eta^2 \|g_t|_{\widetilde{S}_t}\|^2 + 2\eta \langle g_t - \nabla \mathcal{L}(\theta_{t-1}), \theta_* - \theta_{t-1} \rangle \right)$$

$$\leq \frac{2s_*}{s + s_*} \left( 2\|\theta_{t-1} - \theta_*\|^2 + 2\eta \langle \nabla \mathcal{L}(\theta_{t-1}), \theta_* - \theta_{t-1} \rangle + \eta^2 \|g_t|_{\widetilde{S}_t}\|^2 \right)$$

$$\quad + \frac{2s_*}{s + s_*} \eta^2 \|g_t|_{S_{t-1} \cup S_*} - \nabla_{S_{t-1} \cup S_*} \mathcal{L}(\theta_{t-1})\|^2$$

$$\leq \frac{2s_*}{s + s_*} \left( 2\|\theta_{t-1} - \theta_*\|^2 - 2\eta (\mathcal{L}(\theta_{t-1}) - \mathcal{L}(\theta_*)) + \eta^2 \|g_t|_{\widetilde{S}_t}\|^2 \right) + 2s_* \eta^2 \|g_t - \nabla \mathcal{L}(\theta_{t-1})\|_\infty^2$$

$$\leq \frac{2s_*}{s + s_*} \left( 2\|\theta_{t-1} - \theta_*\|^2 + \eta^2 \|g_t|_{\widetilde{S}_t}\|^2 \right) + 2s_* \eta^2 \|g_t - \nabla \mathcal{L}(\theta_{t-1})\|_\infty^2.$$

Here, the second inequality is due to Cauchy-Schwarz inequality and Young's inequality. The third inequality follows from the convexity of $\mathcal{L}$ and the fact that $|S_{t-1} \cup S_*| \leq s + s_*$. The last inequality is due to the optimality $\theta_*$.

Combining (2) with this inequality results in

$$\mathcal{L}(\theta_t) - \mathcal{L}(\theta_{t-1}) \leq \frac{4s_*}{s + s_*} \left( \frac{1}{4\eta} + \frac{L_s}{2} \right) \|\theta_{t-1} - \theta_*\|^2 + \frac{2s_*}{s + s_*} \left( \frac{1}{4} + \frac{\eta L_s}{2} \right) \eta \|g_t|_{\widetilde{S}_t}\|^2$$

$$\quad - \left( \frac{1}{2} + \eta L_s \right) \frac{\eta}{2} \|g_t|_{S_{t-1} \cup S_*}\|^2 - \left( \frac{1}{2} - \eta L_s \right) \frac{\eta}{2} \|g_t|_{S_t \cup S_{t-1}}\|^2$$

$$\quad + 2 \left( s + \left( \frac{1}{4} + \frac{\eta L_s}{2} \right) s_* \right) \eta \|g_t - \nabla \mathcal{L}(\theta_{t-1})\|_\infty^2.$$

By choosing appropriate $s$ and $\eta$ so that $\frac{1}{4} - \frac{\eta L_s}{2} - \frac{2s_*}{s+s_*} \left( \frac{1}{4} + \frac{\eta L_s}{2} \right) \geq 0$, we have

$$\mathcal{L}(\theta_t) - \mathcal{L}(\theta_{t-1}) \leq \frac{4s_*}{s + s_*} \left( \frac{1}{4\eta} + \frac{L_s}{2} \right) \|\theta_{t-1} - \theta_*\|^2$$

$$\quad - \left( 1 - \frac{2s_*}{s + s_*} \right) \left( \frac{1}{4} + \frac{\eta L_s}{2} \right) \eta \|g_t|_{S_{t-1} \cup S_*}\|^2$$

$$\quad + 2 \left( s + \left( \frac{1}{4} + \frac{\eta L_s}{2} \right) s_* \right) \eta \|g_t - \nabla \mathcal{L}(\theta_{t-1})\|_\infty^2.$$

Since $\|\nabla_{S_{t-1} \cup S_*} \mathcal{L}(\theta_{t-1})\|^2 \leq 2\|g_t|_{S_{t-1} \cup S_*}\|^2 + 2\|g_t|_{S_{t-1} \cup S_*} - \nabla_{S_{t-1} \cup S_*} \mathcal{L}(\theta_{t-1})\|^2 \leq 2\|g_t|_{S_{t-1} \cup S_*}\|^2 + 2(s + s_*)\|g_t - \nabla \mathcal{L}(\theta_{t-1})\|_\infty^2$, we have

$$- \left( 1 - \frac{2s_*}{s + s_*} \right) \left( \frac{1}{4} + \frac{\eta L_s}{2} \right) \eta \|g_t|_{S_{t-1} \cup S_*}\|^2$$

$$\leq - \frac{1}{2} \left( 1 - \frac{2s_*}{s + s_*} \right) \left( \frac{1}{4} + \frac{\eta L_s}{2} \right) \eta \|\nabla_{S_{t-1} \cup S_*} \mathcal{L}(\theta_{t-1})\|^2$$

$$\quad + (s - s_*) \left( \frac{1}{4} + \frac{\eta L_s}{2} \right) \eta \|g_t - \nabla \mathcal{L}(\theta_{t-1})\|_\infty^2.$$

Using this inequality, we get

$$
\begin{aligned}
\mathcal{L}(\theta_t) - \mathcal{L}(\theta_{t-1}) \leq\ & \frac{4s_*}{s+s_*}\left(\frac{1}{4\eta} + \frac{L_s}{2}\right)\|\theta_{t-1} - \theta_*\|^2 \\
& - \frac{1}{2}\left(1 - \frac{2s_*}{s+s_*}\right)\left(\frac{1}{4} + \frac{\eta L_s}{2}\right)\eta\|\nabla_{S_{t-1}\cup S_*}\mathcal{L}(\theta_{t-1})\|^2 \\
& + \left(\left(\frac{9}{4} + \frac{\eta L_s}{2}\right)s + \left(\frac{1}{4} + \frac{\eta L_s}{2}\right)s_*\right)\eta\|g_t - \nabla\mathcal{L}(\theta_{t-1})\|_\infty^2 \\
\leq\ & \frac{4s_*}{s+s_*}\left(\frac{1}{4\eta} + \frac{L_s}{2}\right)\|\theta_{t-1} - \theta_*\|^2 \\
& - \frac{1}{2}\left(1 - \frac{2s_*}{s+s_*}\right)\left(\frac{1}{4} + \frac{\eta L_s}{2}\right)\eta\|\nabla_{S_{t-1}\cup S_*}\mathcal{L}(\theta_{t-1})\|^2 \qquad (4)\\
& + s\left(\frac{5}{2} + \eta L_s\right)\eta\|g_t - \nabla\mathcal{L}(\theta_{t-1})\|_\infty^2.
\end{aligned}
$$

Here the last inequality is due to the fact that $s \geq s_*$.

Next we bound the term $\|g_t - \nabla\mathcal{L}(\theta_{t-1})\|_\infty^2$. Observe that we can rewrite

$$
g_t|_j = \frac{1}{B_t}\sum_{b=1}^{B_t}\ell'_{y^{(b)}_{\lceil\frac{j}{s'-s}\rceil}}(\theta_{t-1}^\top x^{(b)}_{\lceil\frac{j}{s'-s}\rceil})x^{(b)}_{\lceil\frac{j}{s-s}\rceil}|_j
$$

for $j \in [d]$. Hence we have

$$
\begin{aligned}
|g_{t,j} - \nabla_j\mathcal{L}(\theta_{t-1})| &= \left|\frac{1}{B_t}\sum_{b=1}^{B_t}\left(\ell'_{y^{(b)}_{\lceil\frac{j}{s'-s}\rceil}}(\theta_{t-1}^\top x^{(b)}_{\lceil\frac{j}{s'-s}\rceil})x^{(b)}_{\lceil\frac{j}{s-s}\rceil}|_j - \mathbb{E}_{(x,y)\sim D}[\ell'_y(\theta_{t-1}^\top x)x|_j]\right)\right| \\
&= \left|\frac{1}{B_t}\sum_{b=1}^{B_t}\left(2(\theta_{t-1}-\theta_*)^\top x^{(b)}_{\lceil\frac{j}{s'-s}\rceil}x^{(b)}_{\lceil\frac{j}{s'-s}\rceil}|_j + 2\xi^{(b)}_{\lceil\frac{j}{s'-s}\rceil}x^{(b)}_{\lceil\frac{j}{s'-s}\rceil}|_j\right.\right. \\
&\qquad\left.\left. - \mathbb{E}_{x\sim D_X}[2(\theta_{t-1}-\theta_*)^\top xx|_j]\right)\right| \\
&\leq 2\left|\frac{1}{B_t}\sum_{b=1}^{B_t}(\theta_{t-1}-\theta_*)^\top x^{(b)}_{\lceil\frac{j}{s'-s}\rceil}x^{(b)}_{\lceil\frac{j}{s'-s}\rceil}|_j - \mathbb{E}_{x\sim D_X}[(\theta_{t-1}-\theta_*)^\top xx|_j]\right| \\
&\quad + 2\left|\frac{1}{B_t}\sum_{b=1}^{B_t}\xi^{(b)}_{\lceil\frac{j}{s'-s}\rceil}x^{(b)}_{\lceil\frac{j}{s'-s}\rceil}|_j\right| \\
&= 2\left|\frac{1}{B_t}\sum_{b=1}^{B_t}X_{b,j,t}\right| + 2\left|\frac{1}{B_t}\sum_{b=1}^{B_t}Y_{b,j,t}\right|,
\end{aligned}
$$

where $X_{b,j,t} \overset{\text{def}}{=} (\theta_{t-1} - \theta_*)^\top x^{(b)}_{\lceil\frac{j}{s'-s}\rceil}x^{(b)}_{\lceil\frac{j}{s'-s}\rceil}|_j - \mathbb{E}_{x\sim D_X}[(\theta_{t-1} - \theta_*)^\top xx|_j]$ and $Y_{b,j,t} \overset{\text{def}}{=} \xi^{(b)}_{\lceil\frac{j}{s'-s}\rceil}x^{(b)}_{\lceil\frac{j}{s'-s}\rceil}|_j$.

First we bound $\left|\frac{1}{B_t}\sum_{b=1}^{B_t}X_{b,j,t}\right|$. Observe that

$$
\begin{aligned}
\left|(\theta_{t-1} - \theta_*)^\top xx|_j\right| &\leq \left|(\theta_{t-1} - \theta_*)^\top x\right|\|x\|_\infty \\
&\leq \|\theta_{t-1} - \theta_*\|_1\|x\|_\infty^2 \\
&\leq \|\theta_{t-1} - \theta_*\|_1 R_\infty^2 \\
&\leq \sqrt{s + s_*}R_\infty^2\|\theta_{t-1} - \theta_*\|_2.
\end{aligned}
$$

Here the second inequality follows from the Cauchy-Schwarz inequality. The first inequality is due to the assumption $\|x\|_\infty \leq R_\infty$ for $x \sim D_X$ almost surely. The last inequality holds because $|\text{supp}(\theta_{t-1} - \theta_*)| \leq s + s_*$.

From this inequality, we have $|X_{b,j,t}| \le 2\sqrt{s + s_*}R_\infty^2\|\theta_{t-1} - \theta_*\|$. Applying Hoeffding's inequality to $\{X_{b,j,t}\}_{b=1}^{B_t}$, we get

$$P\left(\left|\frac{1}{B_t}\sum_{b=1}^{B_t}X_{b,j,t}\right| \ge t \ \middle|\ \|\theta_{t-1} - \theta_*\|^2 = r\right) \le 2\exp\left(-\frac{B_t t^2}{2(s + s_*)R_\infty^4 r}\right)$$

for every $t > 0$. Using union bound property for $j \in [1, d]$, this implies

$$P\left(\forall j \in [1, d] : \left|\frac{1}{B_t}\sum_{b=1}^{B_t}X_{b,j,t}\right| \le \sqrt{\frac{2(s + s_*)R_\infty^4 \log\left(\frac{2d}{\delta_t}\right)}{B_t}r} \ \middle|\ \|\theta_{t-1} - \theta_*\|^2 = r\right) \ge 1 - \delta_t$$

for any $\delta_t > 0$. Since this bound holds for any specific value of $\|\theta_{t-1} - \theta_*\|^2$, we have

$$P\left(\forall j \in [1, d] : \left|\frac{1}{B_t}\sum_{b=1}^{B_t}X_{b,j,t}\right| \le \sqrt{\frac{2(s + s_*)R_\infty^4 \log\left(\frac{2d}{\delta_t}\right)}{B_t}\|\theta_{t-1} - \theta_*\|^2}\right) \ge 1 - \delta_t$$

for any $\delta_t > 0$.

Next we bound $\left|\frac{1}{B_t}\sum_{b=1}^{B_t}Y_{b,j,t}\right|$. Note that given $\left\{x_{\lceil\frac{j}{s'-s}\rceil}^{(b)}|_j\right\}_{b=1}^{B_t}$, $\{Y_{b,j,t}\}_{b=1}^{B_t}$ is a set of independent mean zero sub-gaussian random variables with parameter $x_{\lceil\frac{j}{s'-s}\rceil}^{(b)}|_j^2\sigma^2 \le R_\infty^2\sigma^2$. Hence we have

$$P\left(\left|\frac{1}{B_t}\sum_{b=1}^{B_t}Y_{b,j,t}\right| \ge t \ \middle|\ \left\{x_{\lceil\frac{j}{s'-s}\rceil}^{(b)}|_j\right\}_{b=1}^{B_t}\right) \le \exp\left(-\frac{B_t t^2}{2R_\infty^2\sigma^2}\right).$$

Since the right-hand-side of the above inequality is not dependent on any specific values of $\left\{x_{\lceil\frac{j}{s}\rceil}^{(b)}|_j\right\}_{b=1}^{B_t}$, we can conclude

$$P\left(\left|\frac{1}{B_t}\sum_{b=1}^{B_t}Y_{b,j,t}\right| \ge t\right) \le 2\exp\left(-\frac{B_t t^2}{2R_\infty^2\sigma^2}\right).$$

This gives

$$P\left(\forall j \in [1, d] : \left|\frac{1}{B_t}\sum_{b=1}^{B_t}Y_{b,j,t}\right| \le \sqrt{\frac{2R_\infty^2\sigma^2\log\left(\frac{d}{\delta_t}\right)}{B_t}}\right) \ge 1 - \delta_t$$

for any $\delta_t > 0$.

Combing these results yields

$$P\left(\|g_t - \nabla\mathcal{L}(\theta_{t-1})\|_\infty^2 \le \frac{4(s + s_*)R_\infty^4 \log\left(\frac{2d}{\delta_t}\right)}{B_t}\|\theta_{t-1} - \theta_*\|^2 + \frac{4\sigma^2 R_\infty^2 \log\left(\frac{d}{\delta_t}\right)}{B_t}\right) \ge 1 - 2\delta_t$$

for any $\delta_t > 0$.

Applying this inequality to (4), it holds that

$$\mathcal{L}(\theta_t) - \mathcal{L}(\theta_{t-1}) \le \left(\frac{4s_*}{s + s_*}\left(\frac{1}{4\eta} + \frac{L_s}{2}\right) + 4s(s + s_*)\frac{\left(\frac{5}{2} + \eta L_s\right)R_\infty^4\eta\log\left(\frac{2d}{\delta_t}\right)}{B_t}\right)\|\theta_{t-1} - \theta_*\|^2$$

$$- \frac{1}{2}\left(1 - \frac{2s_*}{s + s_*}\right)\left(\frac{1}{4} + \frac{\eta L_s}{2}\right)\eta\|\nabla_{S_{t-1}\cup S_*}\mathcal{L}(\theta_{t-1})\|^2$$

$$+ 4\sigma^2 s\frac{\left(\frac{5}{2} + \eta L_s\right)R_\infty^2\eta\log\left(\frac{d}{\delta_t}\right)}{B_t}$$

with probability $\geq 1 - 2\delta_t$.

By selecting appropriate $s$, $B_t$ and $\eta$ so that $\frac{4s_*}{s+s_*}\left(\frac{1}{4\eta} + \frac{L_s}{2}\right) + 4s(s+s_*)\frac{\left(\frac{5}{2}+\eta L_s\right)R_\infty^4 \eta\log\left(\frac{2d}{\delta_t}\right)}{B_t} \leq$
$\frac{1}{2}\left(1 - \frac{2s_*}{s+s_*}\right)\left(\frac{1}{4} + \frac{\eta L_s}{2}\right)\eta\frac{\mu_s^2}{4}$, we have

$$\mathcal{L}(\theta_t) - \mathcal{L}(\theta_{t-1}) \leq \frac{1}{2}\left(1 - \frac{2s_*}{s+s_*}\right)\left(\frac{1}{4} + \frac{\eta L_s}{2}\right)\eta\left(\frac{\mu_s^2}{4}\|\theta_{t-1} - \theta_*\|^2 - \|\nabla_{S_{t-1}\cup S_*}\mathcal{L}(\theta_{t-1})\|^2\right)$$
$$+ 4\sigma^2 s\frac{\left(\frac{5}{2} + \eta L_s\right)\eta\log\left(\frac{d}{\delta_t}\right)}{B_t}$$

(5)

with probability $\geq 1 - 2\delta_t$.

To bound the term $\frac{\mu_s^2}{4}\|\theta_{t-1} - \theta_*\|^2 - \|\nabla_{S_{t-1}\cup S_*}\mathcal{L}(\theta_{t-1})\|^2$, we need the following lemma.

*Lemma* A.5. *For any $\theta$ and $\theta_*$ such that $|\mathrm{supp}(\theta)|, |\mathrm{supp}(\theta_*)| \leq s$, it follows that*

$$\frac{\mu_s^2}{4}\|\theta - \theta_*\|^2 - \|\nabla_{S\cup S_*}\mathcal{L}(\theta)\|^2 \leq \mu_s(\mathcal{L}(\theta_*) - \mathcal{L}(\theta)),$$

*where $S = \mathrm{supp}(\theta)$ and $S_* = \mathrm{supp}(\theta_*)$.*

*Proof.* From the restricted strong convexity of $\mathcal{L}$, we have

$$\mathcal{L}(\theta) - \mathcal{L}(\theta_*) \leq \langle\nabla\mathcal{L}(\theta), \theta - \theta_*\rangle - \frac{\mu_s}{2}\|\theta - \theta_*\|^2$$

$$= \langle\nabla_{S\cup S_*}\mathcal{L}(\theta), \theta - \theta_*\rangle - \frac{\mu_s}{2}\|\theta - \theta_*\|^2$$

$$\leq \frac{1}{\mu_s}\|\nabla_{S\cup S_*}\mathcal{L}(\theta)\|^2 - \frac{\mu_s}{4}\|\theta - \theta_*\|^2.$$

Here the last inequality is due to the Cauchy-Schwarz inequality and Young's inequality. This immediately implies the desired inequality. $\square$

Applying Lemma A.5 to (5), we obtain

$$\mathcal{L}(\theta_t) - \mathcal{L}(\theta_{t-1}) \leq \frac{1}{2}\left(1 - \frac{2s_*}{s+s_*}\right)\mu_s\left(\frac{1}{4} + \frac{\eta L_s}{2}\right)\eta(\mathcal{L}(\theta_*) - \mathcal{L}(\theta_{t-1}))$$
$$+ 4\sigma^2 s\frac{\left(\frac{5}{2} + \eta L_s\right)R_\infty^2 \eta\log\left(\frac{d}{\delta_t}\right)}{B_t},$$

and rearranging this inequality results in

$$\mathcal{L}(\theta_t) - \mathcal{L}(\theta_*) \leq (1-\alpha)(\mathcal{L}(\theta_{t-1}) - \mathcal{L}(\theta_*)) + c_t$$

with probability $\geq 1 - 2\delta_t$, where $\alpha = \frac{1}{2}\left(1 - \frac{2s_*}{s+s_*}\right)\mu_s\left(\frac{1}{4} + \frac{\eta L_s}{2}\right)\eta$ and $c_t = 4\sigma^2 s\frac{\left(\frac{5}{2}+\eta L_s\right)R_\infty^2 \eta\log\left(\frac{d}{\delta_t}\right)}{B_t}$.

**Parameters choice**

In the above argument, we have assumed the following conditions:

$$\begin{cases} \frac{1}{2} - \eta L_s > 0, \\ \frac{1}{4} - \frac{\eta L_s}{2} - \frac{2s_*}{s+s_*}\left(\frac{1}{4} + \frac{\eta L_s}{2}\right) \geq 0, \\ \frac{4s_*}{s+s_*}\left(\frac{1}{4\eta} + \frac{L_s}{2}\right) + 4s(s+s_*)\frac{\left(\frac{5}{2}+\eta L_s\right)R_\infty^4 \eta\log\left(\frac{2d}{\delta_t}\right)}{B_t} \leq \frac{1}{2}\left(1 - \frac{2s_*}{s+s_*}\right)\left(\frac{1}{4} + \frac{\eta L_s}{2}\right)\eta\frac{\mu_s^2}{4}. \end{cases}$$

These conditions are satisfied by choosing

$$\begin{cases} \eta = \eta < \frac{1}{2L_s}, \\ B_t \geq \frac{4s}{s_*}(s+s_*)^2\frac{\left(\frac{5}{2}+\eta L_s\right)R_\infty^4 \eta\log\left(\frac{2d}{\delta_t}\right)}{\left(\frac{1}{4\eta} + \frac{L_s}{2}\right)}, \\ s \geq \max\left\{2\left(\frac{\frac{1}{4}+\frac{\eta L_s}{2}}{\frac{1}{4}-\frac{\eta L_s}{2}}\right) - 1, \frac{64}{\eta^2\mu^2} + 1\right\}s_*, \end{cases}$$

for example. $\square$

*Proof of Theorem 4.1.* Let $\eta = \frac{1}{4L_s} = \Theta\left(\frac{1}{L_s}\right)$, $s \geq \max\left\{\left(2\left(\frac{\frac{1}{4}+\frac{\eta L_s}{2}}{\frac{1}{4}-\frac{\eta L_s}{2}}\right)-1\right), \frac{64}{\eta^2\mu_s^2}+1\right\}$ $s_* = O\left(\kappa_s^2 s_*\right)$, $B_t = \left\lceil \max\left\{\frac{4s}{s_*}(s+s_*)^2\frac{\left(\frac{5}{2}+\eta L_s\right)R_\infty^4\eta\log\left(\frac{2d}{\delta_t}\right)}{\left(\frac{1}{4\eta}+\frac{L_s}{2}\right)}, 4\sigma^2 s\frac{\left(\frac{5}{2}+\eta L_s\right)R_\infty^2\eta\log\left(\frac{d}{\delta_t}\right)}{\Delta}\frac{T}{(1-\check\alpha)^T}\right\}\right\rceil =$

$O\left(\frac{\log\left(\frac{Td}{\delta}\right)}{\kappa_s^2}\frac{R_\infty^4}{L_s^2}s^2 \vee \frac{\sigma^2 R_\infty^2\log\left(\frac{Td}{\delta}\right)}{L_s\Delta}\frac{Ts}{(1-\check\alpha)^T}\right) = O\left(\left(\kappa_s^2\frac{R_\infty^4}{L_s^2}\log\left(\frac{Td}{\delta}\right)\right)s^2 \vee \frac{\sigma^2 R_\infty^2\log\left(\frac{Td}{\delta}\right)}{L_s\Delta}\frac{Ts}{(1-\check\alpha)^T}\right)$

and $\delta_t = \frac{1}{2T}\delta$. From Proposition 4.1, we have

$$\mathcal{L}(\theta_t) - \mathcal{L}(\theta_*) \leq (1-\check\alpha)(\mathcal{L}(\theta_{t-1}) - \mathcal{L}(\theta_*)) + \frac{c_t}{B_t}$$

with probability $\geq 1 - 2\delta_t$, where $\check\alpha = \frac{1}{32\kappa_s}$. Recursively using this inequality and applying union bound to the resulting inequality yield

$$\mathcal{L}(\theta_T) - \mathcal{L}(\theta_*) \leq (1-\check\alpha)^T(\mathcal{L}(\theta_0) - \mathcal{L}(\theta_*) + \Delta)$$

with probability $\geq 1 - \delta$. This gives the desired result. $\qquad\square$

## A.2   Proof of Corollary 4.2

*Proof of Corollary 4.2.* From Theorem 4.1, the necessary number of observed samples to achieve $P(\mathcal{L}(\theta_T) - \mathcal{L}(\theta_*) \leq \varepsilon) \geq 1 - \delta$ is given by

$$O\left(\sum_{t=1}^T \frac{d}{s'-s}B_t\right) = O\left(\frac{d}{s'-s}\sum_{t=1}^T\left(\kappa_s^2\frac{R_\infty^4}{L_s^2}\log\left(\frac{dt}{\delta}\right)\right)s^2 + \frac{d}{s'-s}\sum_{t=1}^T\frac{\sigma^2\log\left(\frac{dt}{\delta}\right)}{\mathcal{L}(\theta_0)-\mathcal{L}(\theta_*)}\frac{R_\infty^2}{L_s}\frac{Ts}{(1-\check\alpha)^T}\right)$$

$$\leq O\left(\frac{d}{s'-s}T\left(\kappa_s^2\frac{R_\infty^4}{L_s^2}\log\left(\frac{dT}{\delta}\right)\right)s^2 + \frac{d}{s'-s}\frac{\sigma^2\log\left(\frac{dT}{\delta}\right)}{\mathcal{L}(\theta_0)-\mathcal{L}(\theta_*)}\frac{R_\infty^2}{L_s}\frac{T^2 s}{(1-\check\alpha)^T}\right)$$

$$= O\left(\left(\kappa_s^3\log\left(\frac{d\kappa_s}{\delta}\log\left(\frac{\mathcal{L}(\theta_0)-\mathcal{L}(\theta_*)}{\varepsilon}\right)\right)\log\left(\frac{\mathcal{L}(\theta_0)-\mathcal{L}(\theta_*)}{\varepsilon}\right)\right)\frac{R_\infty^4}{L_s^2}\frac{ds^2}{s'-s}\right.$$
$$\left. + \frac{\sigma^2\kappa_s^2\log\left(\frac{d\kappa_s}{\delta}\log\left(\frac{\mathcal{L}(\theta_0)-\mathcal{L}(\theta_*)}{\varepsilon}\right)\right)\log^2\left(\frac{\mathcal{L}(\theta_0)-\mathcal{L}(\theta_*)}{\varepsilon}\right)}{\mathcal{L}(\theta_0)-\mathcal{L}(\theta_*)}\frac{R_\infty^2}{L_s}\frac{1}{(1-\check\alpha)^T}\frac{ds}{s'-s}\right)$$

$$= O\left(\left(\kappa_s^3\log\left(\frac{d\kappa_s}{\delta}\log\left(\frac{\mathcal{L}(\theta_0)-\mathcal{L}(\theta_*)}{\varepsilon}\right)\right)\log\left(\frac{\mathcal{L}(\theta_0)-\mathcal{L}(\theta_*)}{\varepsilon}\right)\right)\frac{R_\infty^4}{L_s^2}\frac{ds^2}{s'-s}\right.$$
$$+\sigma^2\kappa_s^2\log\left(\frac{d\kappa_s}{\delta}\log\left(\frac{\mathcal{L}(\theta_0)-\mathcal{L}(\theta_*)}{\varepsilon}\right)\right)$$
$$\left.\times\log^2\left(\frac{\mathcal{L}(\theta_0)-\mathcal{L}(\theta_*)}{\varepsilon}\right)\frac{R_\infty^2}{L_s}\frac{ds}{(s'-s)\varepsilon}\right)$$

$$= \widetilde{O}\left(\kappa_s^3\frac{R_\infty^4}{L_s^2}\frac{ds^2}{s'-s} + \sigma^2\kappa_s^2\frac{R_\infty^2}{L_s}\frac{ds}{(s'-s)\varepsilon}\right)$$

$$= \widetilde{O}\left(\frac{\kappa_s R_\infty^4}{\mu_s^2}\frac{d}{s'-s}s^2 + \frac{\kappa_s R_\infty^2}{\mu_s}\frac{d}{s'-s}\frac{\sigma^2 s}{\varepsilon}\right).$$

This is the desired result. $\qquad\square$

# B   Analysis of Algorithm 2

Here, we provide the proof of Theorem 4.3.

*Proof of Theorem 4.3.* Let $\eta = \frac{1}{4L_s}$ and $S = \text{supp}(\theta_0)$. Since $s \geq |\text{supp}(\theta_0)| \vee |\text{supp}(\theta_*)|$, by the restricted smoothness of $\mathcal{L}$, we have

$$
\begin{aligned}
\mathcal{L}(\theta_t) - \mathcal{L}(\theta_{t-1}) &\leq \langle \nabla \mathcal{L}(\theta_{t-1}), \theta_t - \theta_{t-1} \rangle + \frac{L_s}{2} \|\theta_t - \theta_{t-1}\|^2 \\
&= \langle g_t, \theta_t - \theta_{t-1} \rangle + \frac{L_s}{2} \|\theta_t - \theta_{t-1}\|^2 - \langle g_t - \nabla \mathcal{L}(\theta_{t-1}), \theta_t - \theta_{t-1} \rangle \\
&\leq \langle g_t, \theta_t - \theta_{t-1} \rangle + \frac{L_s}{2} \|\theta_t - \theta_{t-1}\|^2 + \frac{\eta}{2} \|g_t|_S - \nabla_S \mathcal{L}(\theta_{t-1})\|^2 + \frac{1}{2\eta} \|\theta_t - \theta_{t-1}\|^2 \\
&= -\frac{\eta}{2}(1 - \eta L_s) \|g_t|_S\|^2 + \frac{\eta}{2} \|g_t|_S - \nabla_S \mathcal{L}(\theta_{t-1})\|^2.
\end{aligned}
$$
(6)

Also, using the argument of bounding $\|g_t - \nabla \mathcal{L}(\theta_{t-1})\|_\infty^2$ in the proof of Proposition A.1, we can show that

$$
P\left( \|g_t|_S - \nabla_S \mathcal{L}(\theta_{t-1})\|^2 \leq \frac{4s(s+s_*)R_\infty^4 \log\left(\frac{2s}{\delta_t}\right)}{B_t} \|\theta_{t-1} - \theta_*\|^2 + \frac{4\sigma^2 s R_\infty^2 \log\left(\frac{s}{\delta_t}\right)}{B_t} \right) \geq 1 - 2\delta_t
$$
(7)

for any $\delta_t > 0$.

**Special Case $S_* \subset S$:**

From (6) and (7), we have

$$
\begin{aligned}
\mathcal{L}(\theta_t) - \mathcal{L}(\theta_{t-1}) &\leq \frac{2s(s+s_*)R_\infty^4 \eta \log\left(\frac{2s}{\delta_t}\right)}{B_t} \|\theta_{t-1} - \theta_*\|^2 - \frac{\eta}{4}(1 - \eta L_s) \|\nabla_S \mathcal{L}(\theta_{t-1})\|^2 \\
&\quad + \frac{2\sigma^2 s R_\infty^2 \eta (2 - \eta L_s) \log\left(\frac{s}{\delta_t}\right)}{B_t},
\end{aligned}
$$

with probability $\geq 1 - 2\delta_t$.

Suppose that $B_t \geq 2s(s+s_*)R_\infty^4 \eta \log\left(\frac{2s}{\delta_t}\right) \frac{4}{\eta(1-\eta L_s)} \frac{4}{\mu_s^2}$. Applying Lemma A.5 to this inequality yields

$$
\mathcal{L}(\theta_t) - \mathcal{L}(\theta_{t-1}) \leq \mu_s \frac{\eta}{4}(1 - \eta L_s)(\mathcal{L}(\theta_*) - \mathcal{L}(\theta_{t-1})) + \frac{2\sigma^2 s R_\infty^2 \eta (2 - \eta L_s) \log\left(\frac{s}{\delta_t}\right)}{B_t}
$$

with probability $\geq 1 - 2\delta_t$. Rearranging this inequality gives

$$
\mathcal{L}(\theta_t) - \mathcal{L}(\theta_*) \leq (1 - \check{\alpha})(\mathcal{L}(\theta_{t-1}) - \mathcal{L}(\theta_*)) + \frac{c_t}{B_t},
$$

where $\check{\alpha} = \frac{1}{32\kappa_s} \leq \mu_s \frac{\eta}{4}(1 - \eta L_s)$ and $c_t = 2\sigma^2 s R_\infty^2 \eta (2 - \eta L_s) \log\left(\frac{s}{\delta_t}\right)$. Therefore setting $B_t \geq \left\lceil \max\left\{ 2s(s+s_*)R_\infty^4 \eta \log\left(\frac{2s}{\delta_t}\right) \frac{4}{\eta(1-\eta L_s)} \frac{4}{\mu_s^2}, \frac{c_t}{\Delta} \frac{T}{(1-\check{\alpha})^T} \right\} \right\rceil = O\left( \frac{R_\infty^4 \log\left(\frac{Ts}{\delta}\right)}{\mu_s^2} s^2 \vee \frac{\sigma^2 R_\infty^2 \log\left(\frac{Ts}{\delta}\right)}{L_s \Delta} \frac{T}{(1-\check{\alpha})^T} s \right)$, where $\delta_t = \frac{1}{2T}\delta$, we obtain

$$
\begin{aligned}
\mathcal{L}(\theta_T) - \mathcal{L}(\theta_*) &\leq (1 - \check{\alpha})^T \left( \mathcal{L}(\theta_0) - \mathcal{L}(\theta_*) + \frac{1}{T}\left( \sum_{t=1}^{T}(1-\check{\alpha})^{t-1} \right)(1-\check{\alpha})^T \Delta \right) \\
&\leq (1 - \check{\alpha})(\mathcal{L}(\theta_0) - \mathcal{L}(\theta_*) + \Delta)
\end{aligned}
$$

with probability $\geq 1 - \delta$.

**General Case:**

Summing (6) from $t = 1$ to $T$, we get

$$\mathcal{L}(\theta_T) - \mathcal{L}(\theta_0) \leq -\frac{1}{2\eta}(1 - \eta L_s) \sum_{t=1}^{T} \|\theta_t - \theta_{t-1}\|^2 + \frac{\eta}{2} \sum_{t=1}^{T} \||g_t|_S - \nabla_S \mathcal{L}(\theta_{t-1})\|^2.$$

Also observe that

$$\begin{aligned}
\|\theta_{t-1} - \theta_*\|^2 &\leq 2\|\theta_{t-1} - \theta_0\|^2 + 2\|\theta_0 - \theta_*\|^2 \\
&\leq t \sum_{t'=1}^{t-1} \|\theta_{t'} - \theta_{t'-1}\|^2 + \frac{4}{\mu_s}(\mathcal{L}(\theta_0) - \mathcal{L}(\theta_*)) \\
&\leq t \sum_{t=1}^{T} \|\theta_t - \theta_{t-1}\|^2 + \frac{4}{\mu_s}(\mathcal{L}(\theta_0) - \mathcal{L}(\theta_*)).
\end{aligned}$$

Using (7), we have

$$\begin{aligned}
\mathcal{L}(\theta_T) - \mathcal{L}(\theta_0) \leq &-\left( \frac{1}{2\eta}(1 - \eta L_s) - \sum_{t=1}^{T} \frac{2ts(s + s_*)R_\infty^4 \eta \log\left(\frac{2s}{\delta_t}\right)}{B_t} \right) \sum_{t=1}^{T} \|\theta_t - \theta_{t-1}\|^2 \\
&+ \sum_{t=1}^{T} \left( \frac{16s(s + s_*)R_\infty^4 \eta \log\left(\frac{2s}{\delta_t}\right)}{\mu_s B_t}(\mathcal{L}(\theta_0) - \mathcal{L}(\theta_*)) + \frac{2\sigma^2 s R_\infty^2 \eta \log\left(\frac{s}{\delta_t}\right)}{B_t} \right)
\end{aligned}$$

Suppose that $B_t \geq \max \left\{ \frac{4ts(s+s_*)R_\infty^4 \eta^2 \log\left(\frac{2s}{\delta_t}\right)}{(1-\eta L_s)}, \frac{16s(s+s_*)R_\infty^4 \eta(1-\check{\alpha})\log\left(\frac{2s}{\delta_t}\right)}{\mu_s}T, \frac{2\sigma^2 s R_\infty^2 \eta \log\left(\frac{s}{\delta_t}\right)}{\Delta}T \right\} = O\left( \frac{R_\infty^4 \log\left(\frac{Ts}{\delta}\right)}{\mu_s^2} Ts^2 \vee \frac{\sigma^2 R_\infty^2 \log\left(\frac{Ts}{\delta}\right)}{L_s \Delta} Ts \right)$, where $\delta_t = \frac{1}{2T}\delta$. Then we obtain

$$\mathcal{L}(\theta_T) - \mathcal{L}(\theta_*) \leq \frac{1}{1 - \check{\alpha}}(\mathcal{L}(\theta_0) - \mathcal{L}(\theta_*)) + \Delta$$

with probability $\geq 1 - \delta$.

Combining the both results and noting that required $B_t$ in the both cases is $O\left( \frac{R_\infty^4 \log\left(\frac{Ts}{\delta}\right)}{\mu_s^2} Ts^2 \vee \frac{\sigma^2 R_\infty^2 \log\left(\frac{Ts}{\delta}\right)}{L_s \Delta} \frac{T}{(1-\check{\alpha})^T} s \right)$ complete the proof. □

## C   Analysis of Algorithm 3

In this section, we give the proofs of Theorem 4.4 and Corollary 4.5.

### C.1   Proof of Theorem 4.4

*Proof of Theorem 4.4.* First we show that Algorithm 4.4 at least achieves the rate of Algorithm 1 in any case.

Combining Theorem 4.1 (with $T = T_k^- = 3$, $\Delta = \Delta_k^- = \frac{1}{2}\check{\alpha}(1 - \check{\alpha})^{k-2}(\mathcal{L}(\widetilde{\theta}_0) - \mathcal{L}(\theta_*))$ and $\delta = \delta_k = \frac{3}{\pi^2 k^2 \delta}$) and Theorem 4.3 (with $T = T_k = \left\lceil \frac{1}{\log((1-\check{\alpha})^{-1})} \log\left( \frac{d}{\Theta(\kappa_s^2)(s'-s)} \vee 1 \right) \right\rceil$, $\Delta =$

$\Delta_k = \frac{1}{2}\check{\alpha}(1-\check{\alpha})^k(\mathcal{L}(\widetilde{\theta}_0) - \mathcal{L}(\theta_*))$ and $\delta = \delta_k = \frac{3}{\pi^2 k^2 \delta}$), we get

$$
\begin{aligned}
\mathcal{L}(\widetilde{\theta}_k) - \mathcal{L}(\theta_*) &\leq \frac{1}{1-\check{\alpha}}(\mathcal{L}(\widetilde{\theta}_k^-) - \mathcal{L}(\theta_*)) + \Delta_k \\
&\leq \frac{1}{1-\check{\alpha}}(1-\check{\alpha})^3(\mathcal{L}(\widetilde{\theta}_{k-1}) - \mathcal{L}(\theta_*) + \Delta_k^-) + \Delta_k \\
&\leq (1-\check{\alpha})^2(\mathcal{L}(\widetilde{\theta}_{k-1}) - \mathcal{L}(\theta_*)) + \check{\alpha}(1-\check{\alpha})^k(\mathcal{L}(\widetilde{\theta}_0) - \mathcal{L}(\theta_*)) \\
&\leq (1-\check{\alpha})^4(\mathcal{L}(\widetilde{\theta}_{k-2}) - \mathcal{L}(\theta_*)) + \check{\alpha}(1-\check{\alpha})^k(1 + (1-\check{\alpha}))(\mathcal{L}(\widetilde{\theta}_0) - \mathcal{L}(\theta_*)) \\
&\leq (1-\check{\alpha})^{2k}(\mathcal{L}(\widetilde{\theta}_0) - \mathcal{L}(\theta_*)) + \check{\alpha}(1-\check{\alpha})^k \sum_{k'=1}^{k}(1-\check{\alpha})^{k'-1}(\mathcal{L}(\widetilde{\theta}_0) - \mathcal{L}(\theta_*)) \\
&\leq 2(1-\check{\alpha})^k(\mathcal{L}(\widetilde{\theta}_0) - \mathcal{L}(\theta_*))
\end{aligned}
\tag{8}
$$

with probability $\geq 1 - \delta$. Hence choosing $K = \frac{1}{\log((1-\check{\alpha})^{-1}))}O\left(\log\left(\frac{\mathcal{L}(\widetilde{\theta}_0) - \mathcal{L}(\theta_*)}{\varepsilon}\right)\right)$ is sufficient for achieving $\mathcal{L}(\widetilde{\theta}_K) - \mathcal{L}(\theta_*) \leq \varepsilon$.

Next we show that Algorithm 4.4 can identifies the support of the optimal solution in finite iterations with high probability.

Applying restricted strong convexity of $\mathcal{L}$ to (8) yields

$$
\|\widetilde{\theta}_k - \theta_*\|^2 \leq \frac{4}{\mu_s}(1-\check{\alpha})^k(\mathcal{L}(\widetilde{\theta}_0) - \mathcal{L}(\theta_*))
$$

for any $k \in \mathbb{N}$. Let $r_{\min} = \min_{j \in \text{supp}(\theta_*)}|\theta_*|_j|$. Assume that there exists $k > \check{k} = \frac{1}{\log((1-\check{\alpha})^{-1}))}\log\left(\frac{4(\mathcal{L}(\widetilde{\theta}_0) - \mathcal{L}(\theta_*))}{r_{\min}^2 \mu_s}\right)$ such that $\text{supp}(\theta_*) \not\subset \text{supp}(\widetilde{\theta}_k)$. We can easily see that $r_{\min}^2 \leq \|\widetilde{\theta}_k - \theta_*\|^2$. Hence we have

$$
r_{\min}^2 \leq \frac{4}{\mu_s}(1-\check{\alpha})^k(\mathcal{L}(\widetilde{\theta}_0) - \mathcal{L}(\theta_*)).
$$

This contradicts the assumption $k > \check{k}$. Hence we conclude that $\text{supp}(\theta_*) \subset \text{supp}(\widetilde{\theta}_k)$ for $k \geq \check{k} + 1$ with probability $\geq 1 - \delta$.

Thus using Theorem 4.3, we have

$$
\begin{aligned}
\mathcal{L}(\widetilde{\theta}_k) - \mathcal{L}(\theta_*) &\leq (1-\check{\alpha})^{T_k}(\mathcal{L}(\widetilde{\theta}_k^-) - \mathcal{L}(\theta_*)) + (1-\check{\alpha})^{T_k}\Delta_k \\
&\leq (1-\check{\alpha})^{T_k + T_k^-}(\mathcal{L}(\widetilde{\theta}_{k-1}) - \mathcal{L}(\theta_*)) + (1-\check{\alpha})^{T_k}((1-\check{\alpha})^{T_k^-}\Delta_k^- + \Delta_k) \\
&\leq (1-\check{\alpha})^{1+\left\lceil\frac{1}{\log((1-\check{\alpha})^{-1}))}\log\left(\frac{d}{\Theta(\kappa_s^2)(s'-s)}\vee 1\right)\right\rceil}(\mathcal{L}(\widetilde{\theta}_{k-1}) - \mathcal{L}(\theta_*)) \\
&\quad + \check{\alpha}(1-\check{\alpha})^{k+\left\lceil\frac{1}{\log((1-\check{\alpha})^{-1}))}\log\left(\frac{d}{\Theta(\kappa_s^2)(s'-s)}\vee 1\right)\right\rceil}(\mathcal{L}(\widetilde{\theta}_0) - \mathcal{L}(\theta_*)) \\
&\leq (1-\check{\alpha})^{(k-\check{k})\left(1+\left\lceil\frac{1}{\log((1-\check{\alpha})^{-1}))}\log\left(\frac{d}{\Theta(\kappa_s^2)(s'-s)}\vee 1\right)\right\rceil\right)}(\mathcal{L}(\widetilde{\theta}_{\check{k}}) - \mathcal{L}(\theta_*)) \\
&\quad + \left\{\check{\alpha}(1-\check{\alpha})^{k+\left\lceil\frac{1}{\log((1-\check{\alpha})^{-1}))}\log\left(\frac{d}{\Theta(\kappa_s^2)(s'-s)}\vee 1\right)\right\rceil}\right. \\
&\qquad \left. \times \sum_{k'=1}^{k-\check{k}}\left((1-\check{\alpha})^{(k'-1)\left(\left\lceil\frac{1}{\log((1-\check{\alpha})^{-1}))}\log\left(\frac{d}{\Theta(\kappa_s^2)(s'-s)}\vee 1\right)\right\rceil\right)}\right)(\mathcal{L}(\widetilde{\theta}_0) - \mathcal{L}(\theta_*))\right\} \\
&\leq 2(1-\check{\alpha})^{k+\left\lceil\frac{1}{\log((1-\check{\alpha})^{-1}))}\log\left(\frac{d}{\Theta(\kappa_s^2)(s'-s)}\vee 1\right)\right\rceil}(\mathcal{L}(\widetilde{\theta}_0) - \mathcal{L}(\theta_*))
\end{aligned}
$$

for every $k \geq \check{k} + 1$ with probability $\geq 1 - 2\delta$.

Therefore running
$$
K = \left\lceil\frac{1}{\log((1-\check{\alpha})^{-1}))}\left(\left(\log\left(\frac{\mathcal{L}(\widetilde{\theta}_0) - \mathcal{L}(\theta_*)}{r_{\min}^2 \mu_s}\right) + \log\left(\frac{(\mathcal{L}(\widetilde{\theta}_0) - \mathcal{L}(\theta_*))(s'-s)}{\check{\alpha}^2 d\varepsilon}\right)\right) \wedge \log\left(\frac{\mathcal{L}(\widetilde{\theta}_0) - \mathcal{L}(\theta_*)}{\varepsilon}\right)\right)\right\rceil =
$$

$$\left\lceil \frac{1}{\log((1-\check{\alpha})^{-1}))} \left( \log\left( \frac{(\mathcal{L}(\widetilde{\theta}_0)-\mathcal{L}(\theta_*))(s'-s)}{r_{\min}^2 \mu_s \check{\alpha}^2 d\varepsilon} \right) \wedge \log\left( \frac{\mathcal{L}(\widetilde{\theta}_0)-\mathcal{L}(\theta_*)}{\varepsilon} \right) \right) \right\rceil$$ iterations is sufficient for achieving $\mathcal{L}(\widetilde{\theta}_K) - \mathcal{L}(\theta_*) \leq \varepsilon$. This completes the proof. $\qquad\square$

## C.2 Proof of Corollary 4.5

*Proof of Corollary 4.5.* We need to bound the number of total observed samples to achieve $\mathcal{L}(\widetilde{\theta}_K) - \mathcal{L}(\theta_*) \leq \varepsilon$. The number of total observed samples is given by

$$O\left( \sum_{k=1}^{K} \left( \sum_{t=1}^{T_k} B_{t,k} + \sum_{t=1}^{T_k^-} \frac{d}{s'-s} B_{t,k}^- \right) \right).$$

$$
\begin{aligned}
O\left( \sum_{k=1}^{K}\sum_{t=1}^{T_k} B_{t,k} \right) &= O\left( \sum_{k=1}^{K}\sum_{t=1}^{T_k} \left( \kappa_s^2 \frac{R_\infty^4}{L_s^2} \log\left( \frac{T_k s}{\delta_k} \right) T_k s^2 + \frac{\sigma^2 \log\left( \frac{T_k s}{\delta_k} \right)}{\Delta_k} \frac{R_\infty^2}{L_s} \frac{T_k}{(1-\check{\alpha})^{T_k}} s \right) \right) \\
&= O\left( \sum_{k=1}^{K} \left( T_k^2 \kappa_s^2 \frac{R_\infty^4}{L_s^2} \log\left( \frac{T_k k^2 s}{\delta} \right) s^2 + \frac{\sigma^2 \log\left( \frac{T_k k^2 s}{\delta} \right)}{\Delta_k} \frac{R_\infty^2}{L_s} \frac{T_k^2}{(1-\check{\alpha})^{T_k}} s \right) \right) \\
&= O\left( \sum_{k=1}^{K} \left( \left\lceil \frac{1}{\log\left((1-\check{\alpha})^{-1}\right)} \log\left( \frac{d}{\Theta(\kappa_s^2)(s'-s)} \vee 1 \right) \right\rceil^2 \right. \right.\\
&\qquad \times \kappa_s^2 \frac{R_\infty^4}{L_s^2} \log\left( \frac{\left\lceil \frac{1}{\log((1-\check{\alpha})^{-1})} \log\left( \frac{d}{\Theta(\kappa_s^2)(s'-s)} \vee 1 \right) \right\rceil k^2 s}{\delta} \right) s^2 \\
&\qquad + \frac{\sigma^2 \log\left( \frac{\left\lceil \frac{1}{\log((1-\check{\alpha})^{-1})} \log\left( \frac{d}{\Theta(\kappa_s^2)(s'-s)} \vee 1 \right) \right\rceil k^2}{\delta} \right)}{\mathcal{L}(\widetilde{\theta}_0) - \mathcal{L}(\theta_*)} \frac{R_\infty^2}{L_s} \\
&\qquad \left. \left. \times \frac{\left\lceil \frac{1}{\log((1-\check{\alpha})^{-1})} \log\left( \frac{d}{\Theta(\kappa_s^2)(s'-s)} \vee 1 \right) \right\rceil^2}{\check{\alpha}(1-\check{\alpha})^{k+\left\lceil \frac{1}{\log((1-\check{\alpha})^{-1})} \log\left( \frac{d}{\Theta(\kappa_s^2)(s'-s)} \vee 1 \right) \right\rceil}} s \right) \right) \\
&= O\left( K \left\lceil \frac{1}{\log\left((1-\check{\alpha})^{-1}\right)} \log \frac{d}{\Theta(\kappa_s^2)(s'-s)} \right\rceil^2 \right.\\
&\qquad \times \kappa_s^2 \frac{R_\infty^4}{L_s^2} \log\left( \frac{\left\lceil \frac{1}{\log((1-\check{\alpha})^{-1})} \log\left( \frac{d}{\Theta(\kappa_s^2)(s'-s)} \vee 1 \right) \right\rceil K^2 s}{\delta} \right) s^2 \\
&\qquad + \frac{\sigma^2 \log\left( \frac{\left\lceil \frac{1}{\log((1-\check{\alpha})^{-1})} \log\left( \frac{d}{\Theta(\kappa_s^2)(s'-s)} \vee 1 \right) \right\rceil K^2 s}{\delta} \right)}{\mathcal{L}(\widetilde{\theta}_0) - \mathcal{L}(\theta_*)} \frac{R_\infty^2}{L_s} \\
&\qquad \left. \times \frac{\left\lceil \frac{1}{\log((1-\check{\alpha})^{-1})} \log\left( \frac{d}{\Theta(\kappa_s^2)(s'-s)} \vee 1 \right) \right\rceil^2}{\check{\alpha}^2(1-\check{\alpha})^{K+\left\lceil \frac{1}{\log((1-\check{\alpha})^{-1})} \log\left( \frac{d}{\Theta(\kappa_s^2)(s'-s)} \vee 1 \right) \right\rceil}} s \right) \\
&= \widetilde{O}\left( K\kappa_s^4 \frac{R_\infty^4}{L_s^2} s^2 + \frac{\sigma^2}{\mathcal{L}(\widetilde{\theta}_0) - \mathcal{L}(\theta_*)} \frac{R_\infty^2}{L_s} \frac{\kappa_s^4}{(1-\check{\alpha})^{K+\left\lceil \frac{1}{\log((1-\check{\alpha})^{-1})} \log\left( \frac{d}{\Theta(\kappa_s^2)(s'-s)} \vee 1 \right) \right\rceil}} s \right).
\end{aligned}
$$

Similarly we have

$$O\left(\sum_{k=1}^{K}\sum_{t=1}^{T_k^-}\frac{d}{s'-s}B_{t,k}^-\right)$$

$$=O\left(\sum_{k=1}^{K}\sum_{t=1}^{T_k^-}\left(\kappa_s^2\frac{R_\infty^4}{L_s^2}\log\left(\frac{td}{\delta_k}\right)\frac{ds^2}{s'-s}+\frac{\sigma^2\log\left(\frac{td}{\delta_k^-}\right)}{\Delta_k^-}\frac{R_\infty^2}{L_s}\frac{t^2}{(1-\check{\alpha})^{T_k^-}}\frac{ds}{s'-s}\right)\right)$$

$$=O\left(\sum_{k=1}^{K}\left(T_k^-\kappa_s^2\frac{R_\infty^4}{L_s^2}\log\left(\frac{T_k^-k^2d}{\delta}\right)\frac{ds^2}{s'-s}+\frac{\sigma^2\log\left(\frac{T_k^-k^2d}{\delta}\right)}{\mathcal{L}(\widetilde{\theta}_0)-\mathcal{L}(\theta_*)}\frac{R_\infty^2}{L_s}\frac{T_k^{-3}}{\check{\alpha}(1-\check{\alpha})^k}\frac{ds}{s'-s}\right)\right)$$

$$=\widetilde{O}\left(K\kappa_s^2\frac{R_\infty^4}{L_s^2}\frac{ds^2}{s'-s}+\frac{\sigma^2}{\mathcal{L}(\widetilde{\theta}_0)-\mathcal{L}(\theta_*)}\frac{R_\infty^2}{L_s}\frac{\kappa_s^2}{(1-\check{\alpha})^K}\frac{ds}{s'-s}\right).$$

Combining these results, we obtain

$$O\left(\sum_{k=1}^{K}\left(\sum_{t=1}^{T_k}B_{t,k}+\sum_{t=1}^{T_k^-}\frac{d}{s'-s}B_{t,k}^-\right)\right)$$

$$=\widetilde{O}\left(K\kappa_s^4\frac{R_\infty^4}{L_s^2}s^2+K\kappa_s^2\frac{R_\infty^4}{L_s^2}\frac{ds^2}{s'-s}\right.$$

$$\left.+\frac{\sigma^2\kappa_s^2}{\mathcal{L}(\widetilde{\theta}_0)-\mathcal{L}(\theta_*)}\frac{R_\infty^2}{L_s}\frac{1}{(1-\check{\alpha})^K}\left(\frac{\kappa_s^2}{(1-\check{\alpha})^{\left\lceil\frac{1}{\log((1-\check{\alpha})^{-1}))}\log\left(\frac{d}{\Theta(\kappa_s^2)(s'-s)}\vee 1\right)\right\rceil}}s+\frac{ds}{s'-s}\right)\right)$$

$$=\widetilde{O}\left(\kappa_s^5\frac{R_\infty^4}{L_s^2}s^2+\kappa_s^3\frac{R_\infty^4}{L_s^2}\frac{ds^2}{s'-s}+\frac{\sigma^2\kappa_s^2}{\mathcal{L}(\widetilde{\theta}_0)-\mathcal{L}(\theta_*)}\frac{R_\infty^2}{L_s}\frac{1}{(1-\check{\alpha})^K}\frac{ds}{s'-s}\right)$$

$$=\widetilde{O}\left(\kappa_s^3\frac{R_\infty^4}{\mu_s^2}s^2+\kappa_s\frac{R_\infty^4}{\mu_s^2}\frac{ds^2}{s'-s}+\sigma^2\kappa_s\frac{R_\infty^2}{\mu_s}\left(\frac{\kappa_s^2s}{\mu_sr_{\min}^2}\wedge\frac{ds}{s'-s}\right)\frac{1}{\varepsilon}\right),$$

which complete the proof. $\qquad\square$

## References

[1] P. Jain, A. Tewari, and P. Kar. On iterative hard thresholding methods for high-dimensional m-estimation. In *Advances in Neural Information Processing Systems*, pages 685–693, 2014.