[Reviews · NeurIPS 2018]

Reviewer 1



This looks an interesting result. Authors suggest a couple of methods, inspired by "mini batch strategy", for solving linear regression when the signal of interest is sparse and one can observe the data partially. The problem and their proposed methods are clearly described in the paper. Theoretical guarantees for convergence of the algorithms are provided which is supported by practical evidence given in the paper. As the authors suggest, these methods look to outperform other existing methods in term of "sample complexity". I think result presentation can be improved by paying attention to the following points : 1) I think hiding effect of noise parameter $ \sigma^2 $ in the result presented in table 1 is not a good idea at all. It shows the robustness of the algorithm and omitting it may cause some confusions. For instance, in the presented result in table 1, it seems just with scaling the problem (sending $ r_min $ to infinity), we can improve the performance of the proposed Hybrid algorithm and make sample complexity independent of $ \epsilon $. It is not the case because scaling blows up the noise level which is not visible in the way that they present the complexity comparison. 2) It seems choice of the parameter $ s $, defined in line 116, plays a crucial role in the performance of the algorithm. However, it is not clear to me how the authors choose this parameter for the practical results. Especially, because the level of sparsity $ s* $ is not known and I assume $ s >= s* $. It is reasonable to assume an upper bound for $ s* $ and tune $ s $ with standard methods, but neither are mentioned in the paper. 3) I think the comparison in the simulation part is not fair because of the following reasons, i) There is no run-time data for the numerical experiments. Sample complexity is not the only factor that shows the complexity of the algorithm. In many problems, including compressive linear regression, it is possible to solve the problem with information theoretic bounds for sample complexity at the cost of exponential run-time. If the claim is having efficient sample complexity, some computational complexity should be included as well. Moreover, in comparison of different methods, this should be considered. ii) I think that Dantzig is performing horribly in the synthetic data but pretty well in the real data can be a sign that synthetic data is chosen to artificially fit the proposed method. 4) The maximum chance that the analysis can guarantee for the success of the Hybrid algorithm, as Theorem 4.4 suggests, is less than 67%, regardless of the number of samples we have. Although the main point of the work is solving the problem with partially finite observations, not having asymptotic consistency is a weakness for the theoretical result. In general, I think this is an interesting result while there is a large room for improvement.

Reviewer 2



The paper presents two new algorithms for solving the sparse linear regression problem, and show that the proposed methods have a better sample complexity compared to the existing methods. Different from the popular RIP or linear independence assumptions, the paper assumes restricted smoothness and restricted strong convexity instead. Just one question: I assume the H_s above line 117 and in Algorithm 1 is a projection, but it seems not defined in the draft?

Reviewer 3



The paper studies an important problem of sparse linear regression where the features at each example are missing or unobserved. The unobserved patterns are different from samples but we are allowed to have a control to what features are selected at each iteration of the algorithm. To find the true underlying sparse parameter, a standard method is to solve the popular risk l2 minimization with sparse constraint. Contribution: The authors propose to solve this sparse constraint optimization problem via stochastic iterative hard thresholding method. This algorithm has been studied before but the authors make a twist to go around the unobserved features issue. They also propose a refining algorithm that can fine tune the solution after the correct support is attained. They additionally provide a hybrid method that combines the searching and fine tuning algorithms in a iterative manner. These algorithms enjoy theoretical guarantee with sample complexity being better that previous methods, e.g. the sample complexity is linearly proportional with data dimension and inversely proportional with the function value error. Main problem: The paper is clear and easy to follow until the algorithm description in paper 4. In particular, it is unclear to me what does x_i^{(b)} mean in the Algorithm 1. Is the subscript i meant an entry of the vector x^{(b)} where b is the sample index. What is x_i^{(b)} | J_i? These notations are not defined anywhere in the paper. For that notation problem, I can only follow to prove up the the equation (4) in the supplement. Up to that point, the proof is pretty standard and follow closely the analysis of the citation [1]. The authors may want to emphasize which part of the analysis is novel and is not seen by previous works. Although I can’t not judge the correctness of the proof, the theorems sound right to me. Overall, the paper offers a good theoretical improvement, its clarity needs to be improved in order to reach to wide audience.